# On Convergence of the Alternating Directions Stochastic Gradient Hamiltonian Monte Carlo (SGHMC) Algorithms

## Abstract

We study convergence rates of practical Hamiltonian Monte Carlo (HMC) style algorithms where the Hamiltonian motion is approximated with leapfrog integration and where gradients of the log target density are accessed via a noisy gradient estimation. Importantly, our analysis extends the class of auxiliary distributions allowed via the novel HMC procedure of alternating directions (AD). The convergence analysis is based on the investigation of the Dirichlet forms associated with the underlying Markov chain driving the algorithms. For this purpose, we provide a detailed analysis on the error of the leapfrog integrator for Hamiltonian motions when both the kinetic and potential energy functions are in general form. We characterize the explicit dependence of the convergence rates on key parameters such as the problem dimension, functional properties of the target and auxiliary distributions and the quality of the noisy gradient estimation. Our analysis also identifies a crucial derivative condition on the log density of the auxiliary distribution, and we show that Gaussians (auxiliaries for standard HMC) as well as common choices of general auxiliaries for ADHMC satisfy this condition.

## 1 Introduction

The Hamiltonian Monte Carlo (HMC) algorithm, which can be traced to the statistical physics literature (Duane et al., 1987; Horowitz, 1991) has recently seen much wider application in modern statistical applications such as Bayesian inference (Carpenter et al., 2017), learning using deep neural networks (Vellenga et al., 2025), knowledge graph completion (Wang et al., 2025) and generative modeling using diffusion models (Singh & Jakob, 2024). Given a *target* distribution which is known to be proportional to a given positive, integrable function $\mathfrak{f}(q)$, $q \in \mathbb{R}^d$, Markov chain Monte Carlo (MCMC) algorithms are employed to provide either estimations of the normalizing constant (known as the partition function especially in statistical physics) to $\mathfrak{f}$ or to generate samples that are (approximately) from the target distribution. In abstract, an MCMC iteration starts with the current iterate and selects a proposed new iterate, usually by sampling from a secondary distribution; it then applies a correction step, typically Metropolis-Hastings (MH) rejection (Metropolis et al., 1953), to ensure reversibility of the chain and convergence to the desired target. HMC, a member of the MCMC family, utilizes the invariance and ergodic properties of Hamiltonian motion to offer additional benefits in performance compared to generic MCMC algorithms. With the help of a user chosen *auxiliary* distribution $\mathfrak{g}(p)$ defined over an additional variable $p \in \mathbb{R}^d$ the algorithm generates its proposal by following dynamics defined by Hamilton's equations that preserve the energy $\mathcal{H}(q, p) = -(\log \mathfrak{f}(q) + \log \mathfrak{g}(p))$ (see Sec. 2.3.1 for details), where $U(q) := -\log \mathfrak{f}(q)$ has historically been called the potential energy and $V(p) := -\log \mathfrak{g}(p)$ the kinetic energy. The chief advantage lies in how well the motion dynamics are implemented; an exact implementation preserves the joint product density even when making large moves and hence does not require any further correction (e.g. an MH rejection step) to ensure proper convergence. In practical instances where the motion is numerically approximated, excellent discrete motion implementations such as the Leapfrog symplectic integrator ensure that rejection probability is low in the necessary MH correction even in high dimensional settings.

The literature focuses on analysis of HMC with Gaussians as auxiliary distribution $\mathfrak{g}(p)$, which corresponds to a straightforward quadratic kinetic energy function. There are several different approaches for both

qualitative and quantitative analysis on the key question of the convergence and performance of the HMC algorithms with Gaussian auxiliaries, see,e.g. Zou & Gu (2021); Li et al. (2019); Gao et al. (2021). As HMC is increasingly used in more complex applications, it becomes desirable that the algorithm as well as its analysis can be extended to allow more flexibility in the selection of the auxiliary distributions. We are motivated by the observation in Ghosh et al. (2025) that the careful selection of a multi-modal mixture of Gaussians as the auxiliary helps speed up convergence to their multi-modal example target distribution. The symmetry of the Gaussian distribution when used as the auxiliary is key to ensuring reversibility of standard HMC. To similarly preserve reversibility while allowing for a general auxiliary, Ghosh et al. (2025) propose a novel algorithm called the Alternating Direction HMC (ADHMC) (see Sec. 2.3.3).

In many practical situations, the gradient of the potential energy $U(q)$ of the target density function, which is essential for the running of HMC, is not available or difficult to compute. For instance, consider the use of MCMC to sample from the Bayesian posterior density given by $\mathfrak{f}(q) \propto \pi(q)\mathsf{P}(x \mid q)$, where $\pi(q)$ is the prior distribution and $\mathsf{P}(x \mid q)$ represents the likelihood of observing data $x$ given likelihood model parameters $q$. In Bayesian AI/ML models, the log likelihood is calculated as $\log \mathsf{P}(x \mid q) = \left(\frac{1}{M}\right) \sum_{m=1}^{M} \log \mathsf{P}(x_m \mid q)$ over an independently sampled dataset $x := \{x_m\}_{m=1}^{M}$. Statistically learning models such as support vector machines (Huq & Cleland, 1990) describe the individual $\log \mathsf{P}(x_m \mid q)$. Practitioners resort to approximation techniques such as mini-batching to estimate the gradient of the log-posterior over large $M$-sized datasets.

In the abstract, one substitutes the exact calculation of the function $\nabla U(q)$ by an (unbiased) estimator $\tilde{G}_r(q, \xi)$ with an independent random variable $\xi$. Generally speaking, the computation complexity for calculating $\tilde{G}_r(q, \xi)$ is considered to be significantly less than that of $\nabla U(q)$, especially in high dimensions. Examples of such estimators include Mini-batch stochastic gradient, Stochastic variance reduced gradient, Stochastic averaged gradient and Control variate gradient, as summarized in Zou & Gu (2021).

Our primary purpose is to present a unified quantitative convergence analysis of the family of Alternating Direction HMC algorithms that allow for general not necessarily symmetric auxiliary distributions satisfying Assumptions 1 and 2. The analysis presented here also allows for noisy (inexpensive) computations that estimate the gradient $\nabla U(q)$ of the potential of the target distribution, and where Hamiltonian motion is implemented with leapfrog symplectic integrators (requiring additional MH correction). We take an analytical approach based on the analysis of the Dirichlet forms that are defined by the underlying Markov chain, and are able to quantitatively characterize convergence rates of Alternating Direction SGHMC under mild conditions on the target density, the auxiliary density and the noisy gradient estimation implementations. These, to the best of our knowledge, are the first set of results on the convergence rates of HMC with this kind of generality.

## 1.1 Literature review.

As a special instance of MCMC, an HMC algorithm is driven by a Markov chain in a general state space. Hence, the analysis of its convergence relies on the analysis of this Markov chain. Convergence of Markov chains, or more general Markov processes, is a central topic in probability theory. The variety of different approaches is larger than what a few books, see, e.g. Meyn & Tweedie (1993) and Levin et al. (2009), can cover. The main approach taken in this paper is based on the Dirichlet form, a systematic treatment of which can be found in Fukushima et al. (2010). Intuitively, this analysis focuses on establishing functional relationships that quantitatively characterize the evolution of the Markov chain, thus facilitating the convergence analysis. These concepts will be introduced in Sec. 2. The main technical portion of the paper will address the estimation of the Dirichlet form.

The research on the convergence and convergence rates of HMC has been concentrated on the case of the auxiliary distribution $\mathfrak{g}(p)$ being a (conditional) Gaussian. Theoretical understanding of geometric convergence have been developed for these cases, via analytical methods (including comparison theorems for differential equations in Chen & Vempala (2019)), or probabilistic methods, such as Harris recurrence techniques Bou-Rabee & Sanz-Serna (2017) and coupling Bou-Rabee et al. (2020). For HMC with general auxiliary distributions, qualitative results are obtained in Ghosh et al. (2022a) and Ghosh et al. (2022b).

Stoltz & Trstanova (2018) have generalized HMC to use non-Gaussian auxiliaries and study the resulting Langevin style continuous flow dynamics. Their chief motivation is to have a flatter kinetic energy around the origin and a steeper than quadratic tail. Their results differ from ours in two key aspects: the convergence of only the continuous flow scheme to the desired target is established, and more importantly they assume the kinetic energy is still symmetric around the origin in order to preserve reversibility of the MC. In contrast, we present a complete convergence analysis of our discretization scheme, and also do not place this symmetry assumption on the auxiliary. Instead, the alternating-direction generalization of HMC takes consecutive steps in forward and backward Hamiltonian motions to preserve reversibility. Ghosh et al. (2025) describe a technique of using multi-modal Gaussian mixtures as the kinetic energy $V(p)$ in order to help accelerate the convergence of HMC. They construct the mixture adaptively by placing modes at the observed cluster points that emerge from the HMC iterates, which are likely locations of modes of the potential function $U(q)$. The empirically observed acceleration is implied to be because the $V(p)$ adaptively adopts the multi-modal characteristics of $U(q)$.

The problem we address is an instance of the general problem of HMC implemented with practical discrete dynamics and a noisy gradient estimation, and there is existing literature on convergence of standard HMC under these conditions, see e.g. Zou & Gu (2021); Li et al. (2019) and Gao et al. (2021). Our analysis in this paper extends this with explicit convergence rate estimates for further general assumptions of noisy gradient estimations and discrete implementation of motion under general auxiliary distributions, for which we study the alternating direction modification to HMC (Ghosh et al., 2025).

The literature also provides extensive quantitative studies on establishing the dependence of convergence rates on parameters of the algorithms, including the dimension $d$ of the underlying space, the function properties of the target distribution and the quality of the numerical integrators. The main candidate for such analysis is the *unadjusted* HMC, where the gradient estimation is stochastic and motion is discretized but the Metropolis-Hastings correction step is dropped; see e.g. Gouraud et al. (2023), Shen & Lee (2019), Cao et al. (2021) and Bou-Rabee & Marsden (2022). This removal is motivated as a technique to sample the global minima of $U(q)$, since the MH step implementation still needs accurate evaluations of the function $U(q)$. Dropping this step however changes the limiting distribution of the MCMC procedure, and so the main results in this stream of work are not on convergence rates but on approximation errors. For instance, Mangoubi & Vishnoi (2018) guarantee that their algorithm, under strong convexity assumption on the target and warm start, obtains approximately close samples with high probability.

Our main focus is the adjusted HMC (that is with MH correction), which guarantees the outcome will eventually converge to the target distribution, as needed in Bayesian posterior sampling for unbiased estimation of characteristics of the target such as expectations. This case is studied in the literature for example in Chen et al. (2020), Beskos et al. (2013) and Chen & Gatmiry (2023), where results are developed on the dependence on $d$ and the "gradient complexity", that is the number of calls needed to the noisy gradient estimation, under various sets of conditions. Our results are in broad agreement with those for standard HMC as outlined below.

### 1.2 Summary of our contributions

**Error Estimation:** We present a detailed and comprehensive analysis in Lemmata 4.1 and 4.2 on the quality of Leapfrog implementations of the symplectic integration for Hamiltonian equations with general kinetic energy. This is the key for the analysis of HMC with general auxiliary distributions and stochastic gradient. It not only serves as the main technical component for convergence results in this paper, but it can also be used as a building block for the analysis of many other variations of HMC, such as ADHMC seen in this paper, as well as other systems where symplectic integrations and gradient estimation are required.

**Convergence:** Quantitative bound on the performance for SG-ADHMC algorithms with general auxiliary distributions are derived. To our best knowledge, this is the first such results with this level of generality. In addition, these bounds are expressed in explicit form containing system parameters including the dimension. We show that geometric convergence in total-variation distance can be achieved by the ADHMC algorithm using Leapfrog integrated Hamiltonian motion, with either exact gradient calculation (Theorem 3.1) or stochastic gradient estimate (Corollary 3.2), at a rate whose dependence on parameters $\eta$ (the motion dis-

cretization step length) is in agreement with those for standard HMC Chen & Gatmiry (2023) and Lee et al. (2020). Moreover, the step-lengths in our analysis also need to be upper bounded by $O(1/d)$. (This dependence can be relaxed to $O(d^{-1/2})$ for standard HMC with Gaussian auxiliary and under strong convexity of $U(q)$ and crucially with a "warm up" assumption.) Our analysis also identifies a crucial derivative condition (Assumption 4) on the kinetic energy of general auxiliary distributions that allows for geometric convergence, and we show that Gaussians (auxiliaries for standard HMC) as well as common choices of general auxiliaries for ADHMC (as studied in Ghosh et al. (2025)) satisfy this condition. Finally, in Propositions 4.1 and 4.2 we derive quantitative characterization of the continuity of the leapfrog implementation of HMC in probability space, which are of independent interest.

**Methods:** The method we used here consists of Dirichlet form and functional inequalities. They offer clearness in concepts and flexibility in analysis, and appear to be promising in achieving both qualitative and quantitative results, and we hope that they would find more applications within this community. We also aim to remove some of the restrictions implied by the assumptions in our results and apply them to more general systems in the future.

### 1.3 Organization

The rest of the paper is organized as follows. In Sec. 2, we introduce the basic HMC algorithm and its various implementations (standard, alternating direction, stochastic gradients etc.), and present the assumptions placed on the functions for our analysis. Geometric convergence is discussed in Sec. 3. Some further ramifications of the various intermediate steps of our analysis will be presented in Sec. 4, and the paper concludes in Sec. 5.

## 2 Algorithms and Assumptions

### 2.1 Definitions, Notations and Assumptions

For any $\mathfrak{q} \in \mathbb{R}^d$, and $p \in \mathbb{Z}_+$, the $p$-norm is defined as $\|q\|_p = (\sum_{i=1}^d q_i^p)^{1/p}$. For a random variable $q$ defined on $\mathbb{R}^d$, define $\||q\||_p := (\mathbb{E}\|q\|_2^p)^{1/p}$. For a $d \times d$ matrix $A$, the operator norm (aka spectral norm) is defined as $\|A\| = \sup_{\|x\|_2=1} \|Ax\|_2$ and Frobenius norm as $\|A\|_F = \sqrt{\sum_{i,j=1}^d A_{ij}^2}$. For any smooth function $f : \mathbb{R}^d \to \mathbb{R}$, $\nabla^3 f$ can be treated as a tensor, and

$$\|\nabla^3 f\| = \sup\left\{ \left| \sum_{i,j,k=1}^d \frac{\partial^3 f}{\partial x_i \partial x_j \partial x_k} u_i v_j w_k \right| : \|u\|_2, \|v\|_2, \|w\|_2 \le 1 \right\}.$$

One of the key assumptions in Ghosh et al. (2022b) under which the *geometric convergence* of HMC is established is the *uniform strongly logarithmic concavity* of both the target and auxiliary distributions. This is equivalent to an assumption on the convexity and derivative-Lipshitzness conditions on the energy functions.

**Definition 1** ($\mathcal{S}_{\ell,L}(\mathbb{R}^d)$ **class**). *A function $W : \mathbb{R}^d \to \mathbb{R}$ is called to be a class $\mathcal{S}_{\ell,L}$ for some $\ell, L > 0$ if the following holds for any $x_1, x_2 \in \mathbb{R}^d$ and $t \in [0,1]$,*

$$W((1-t)x_1 + tx_2) \le (1-t)W(x_1) + tW(x_2) - \frac{\ell}{2}t(1-t)\|x_1 - x_2\|^2,$$

*and $\|\nabla W(x_2) - \nabla W(x_1)\| \le L\|x_2 - x_1\|$.*

**Remark 2.1.** *The function class $\mathcal{S}_{\ell,L}$ is the same as that of $\mathcal{S}_{\ell,L}^{1,1}(\mathbb{R}^d)$ class in Nesterov (2003). It should be easy to see that $\ell \operatorname{Id} \preceq \nabla^2 W \preceq L \operatorname{Id}$, if $\nabla^2 W$ exists. For any two matrices $A$ and $B$, $A \preceq B$ means that $B - A$ is positive semidefinite.*

**Assumption 1.** *There exist $0 < \ell_U \le L_U < \infty$ and $0 < \ell_V \le L_V < \infty$ such that, $U \in \mathcal{S}_{\ell_U,L_U}(\mathbb{R}^d)$, and $V \in \mathcal{S}_{\ell_V,L_V}(\mathbb{R}^d)$.*

**Assumption 2.** *Both $U$ and $V$ have third derivatives, and there exist $0 < T_U, T_V < \infty$ such that $\sup_{q \in \mathbb{R}^d} \|\nabla^3 U(q)\| \le T_U$ and $\sup_{q \in \mathbb{R}^d} \|\nabla^3 V(q)\| \le T_V$.*

## 2.2 Dirichlet Form and Spectral Gap.

Dirichlet form, as a generalization of the Laplace operator, is an important concept in analysis, a systematic treatment of its connection to probability theory, especially the symmetric Markov processes can be found in Fukushima et al. (2010). We here provide its basic definition for completeness.

**Definition 2.** *A symmetric bilinear form $\mathcal{E}(\cdot, \cdot)$ on the Hilbert space $L^2(X, m)$ with $X$ being a metric space and $m$ a Borel measure is* Markovian *if for any $\epsilon > 0$, there exists a real function $\phi_\epsilon(t) : \mathbb{R} \to \mathbb{R}$ satisfying (A) $\phi_\epsilon(t) \in [-\epsilon, 1 + \epsilon], \forall t \in \mathbb{R}$, (B) $\phi_\epsilon(t) = t$ for $t \in [0, 1]$, and (C) $0 \leq \phi_\epsilon(t') - \phi_\epsilon(t) \leq t' - t$ whenever $t < t'$, such that $\mathcal{E}(\phi_\epsilon(u), \phi_\epsilon(u)) \leq \mathcal{E}(u, u)$ for all $u \in L^2(X, m)$ .*

**Definition 3.** *A symmetric bilinear form is a* Dirichlet form *if it is both Markovian and closed.*

For a reversible Markov chain on $\mathbb{R}^d$ with invariant measure $\pi(x)$ and transition kernel $P(x, A)$, such as the ones we consider in this paper, a natural Dirichlet form on $L^2(\mathbb{R}^d, \pi)$ (we will write $L^2$ in the sequel whenever the context is clear) is given by

$$\mathcal{E}(g, h) = \int_X \int_X [g(x) - g(y)][h(x) - h(y)]\pi(dx)P(x, dy).$$

For such a Markov chain, its spectral gap is denoted as $(1 - \lambda_2)$, where $\lambda_2$ represents the second largest eigenvalue of the operator associated with the transition kernel. The spectral gap has the following representation:

$$1 - \lambda_2 = \inf_{h \text{ not constant}} \frac{\mathcal{E}(h, h)}{\mathcal{V}_\pi(h)},$$

where $\mathcal{V}_\pi(h) := \int_X \int_X (h(x) - h(y))^2 \pi(dx)\pi(dy)$. The Dirichlet form approach on the convergence rate is closely related to the study of conductance originated in Cheeger (1969) and carried out by a series of subsequent studies. A detailed exposition of the results and basic arguments can be found in Lawler & Sokal (1988). For Markov chain generated by the HMC algorithm with leapfrog implementation, the availability of the invariant measure (up to a constant) $\mathfrak{f}$ and the explicit form of transition make the Dirichlet form approach very appealing.

## 2.3 Hamiltonian Monte-Carlo Algorithms

### 2.3.1 Basic Algorithms

A generic HMC algorithm, see Algorithm 1, on Euclidean space usually consists of three operations at each step, with a given starting point $q \in \mathbb{R}^d$: (1) "lift", it is also called "spread" in literature, where a sample $p$ is drawn from the auxiliary distribution with density $\mathfrak{g}(\cdot)$, $(q, p)$ will be a point in the (symplectic) space of $\mathbb{R}^{2d}$; (2) "rotation" to a new point, $(\hat{q}, \hat{p})$, is identified by the Hamiltonian trajectory with energy $\mathcal{H}(q, p) = -\log[\mathfrak{f}(q)\mathfrak{g}(p)] = U(q) + V(p)$, represented here by its potential and kinetic components; (3) "projection", $\hat{q}$ will be the starting point of the next step. More specifically, Algorithm 1 presents the standard HMC with Gaussian auxiliaries (kinetic term $V(p) = p^2/2$) and utilizes a noisy gradient estimation $\nabla^\omega U(\cdot)$ for the gradient of the potential $U(q)$. It also uses $K$ steps of the symplectic integrator each with size $\eta$ to implement Hamiltonian motion.

**Leapfrog implementation of the symplectic integration.** The exact Hamiltonian integration to determine the movement from $(q, p)$ to $(\hat{q}, \hat{p})$ is in general expensive to calculate, and the following well-known Leapfrog approximation is considered here.

$$\hat{q} = q + \eta \nabla V \left( p - \frac{1}{2} \eta \nabla U(q) \right), \qquad \hat{p} = p - \eta \frac{\nabla U(q) + \nabla U(\hat{q})}{2}. \tag{1}$$

### 2.3.2 Stochastic Gradient HMC

As discussed in Zou & Gu (2021), while the functions of the auxiliary distribution can be more easily calculated since it is chosen by the users, calculations of gradients of the target density are not always

readily available or can be attained with low costs. Therefore, they have been approximated in practice, for examples, by *Mini-batch stochastic gradient.* In this case, $U(\mathfrak{q}) = \sum_{i=1}^{n} U_i(\mathfrak{q})$, $\xi$ is a random variable follows a discrete uniform distribution where $\mathcal{I}$, a size-$B$ subset of $[n]$, is selected, then the gradient of $U(\mathfrak{q})$ is estimated by the following unbiased estimator, $\tilde{G}_r(\mathfrak{q}, \xi) = \frac{n}{B} \sum_{i \in \mathcal{I}} \nabla U_i(\mathfrak{q})$. Variations of the above method include *stochastic variance reduced gradient*, *stochastic averaged gradient*, and *control variate gradient*. Details can be found in Zou & Gu (2021) and references therein. We start with an assumption on the quality of the noisy gradient estimation.

**Assumption 3.** *At each step of the leapfrog calculation, the term $\nabla U$ is approximated by an independent random variable, denoted as $\nabla U^\omega$. Furthermore, there exist $0 < \underline{\ell} \leq \bar{L} < \infty$ and $\bar{T} > 0$, such that, $U^\omega \in \mathcal{S}_{\ell^\omega, L^\omega}(\mathbb{R}^d)$ almost surely, with $\ell^\omega \geq \underline{\ell} > 0$ and $L^\omega \leq \bar{L} < \infty$ and $\|\nabla^3 U^\omega\| \leq \bar{T}$ almost surely.*

**An Acceptance/Rejection step.** At each step of the HMC implementation, leapfrog procedure 1 will be invoked $K$ time, produce a proposal $(Q_K, P_K)$ from the initial state $(Q_0, P_0)$ formed by the initial position $Q_0$ and $P_0$ sampled from the auxiliary distribution $\mathfrak{g}$. The proposal is then accepted with the probability that, when transformed via log, is seen as

$$\log \min \left\{ 1, \frac{\mathfrak{f}(Q_K)\mathfrak{g}(P_K)}{\mathfrak{f}(Q_0)\mathfrak{g}(P_0)} \right\} = \min \left\{ 0, \mathcal{H}(Q_0, P_0) - \mathcal{H}(Q_K, P_K) \right\}. \tag{2}$$

Thus, if the Hamiltonian motion implementation were exact, there would be no sample rejection.

### 2.3.3 ADHMC

Algorithm 2 presents the Alternating Direction HMC algorithm that can be used with general auxiliaries (with kinetic term $V(p)$). As before, noisy gradient estimation $\nabla U^\omega(q)$ provides an estimate of the gradient of the potential $U(q)$. A modification of standard SGHMC Algorithm is required because the asymmetry of the general auxiliary may not preserve the reversibility of the iterates of the standard algorithm. We recover this by following a procedure that alternates Hamiltonian motion in forward and backward directions for the same length $T$. This modified method is called the Alternating Direction HMC (ADHMC), and is proposed and analyzed in Ghosh et al. (2025). Both sets of Hamiltonian motions are implemented using $K$ steps of the symplectic integrator of size $\eta$ : the first implements forward motion, and the second implements backward motion.

One step of the ADHMC Algorithm 2 for asymmetrical auxiliaries $\mathfrak{g}$ starts from a $Q_0 \in \mathbb{R}^d$ by generating a sample $P_0 \in \mathbb{R}^d$ and applying forward Hamiltonian motion (via the discretized symplectic integrator steps shown in Algorithm 2) for fixed steps $K$ that carries the pair $(Q_0, P_0)$ to a new point $(Q_K, P_K)$. The first coordinate is retained as the intermediate $Q_0'$. Then, another momentum $P_0' \in \mathbb{R}^d$ is sampled and the backward Hamiltonian motion carries $(Q_0', P_0')$ to $(Q_K', P_0')$, yielding the candidate $Q_K'$ for the next state. Similarly, should we start ADHMC with $Q_K'$, the pair of momentum vectors $P_K'$ and $P_K$ will take us back to $Q_0$ through $Q_0' = Q_K$.

Define $\Pi_{q_0}^{-f}(q)$ as the momentum that (uniquely) carries $q_0$ to $q$ via the discrete forward Hamiltonian motion iterations. In other words, $p = \Pi_{q_0}^{-f}(q)$ is such that under forward Hamiltonian iterations, $(q_0, p)$ transforms to $(q, p')$ for some $p'$. Similarly, let $\Pi_{q_0}^{-b}(q)$ denote the momentum that carries $q_0$ to $q$ via the discretized backward Hamiltonian motion iterations. The Algorithm 2 accepts the proposed move to $Q_K'$ with probability

$$\mathcal{P}(Q_0, Q_0', Q_K') = \min \left\{ 1, \frac{\mathfrak{f}(Q_K) \, \mathfrak{g}(\Pi_{Q_K'}^{-f}(Q_0')) \, \mathfrak{g}(\Pi_{Q_0'}^{-b}(Q_0))}{\mathfrak{f}(Q_0) \, \mathfrak{g}(\Pi_{Q_0}^{-f}(Q_0')) \, \mathfrak{g}(\Pi_{Q_0'}^{-b}(Q_K'))} \right\}, \tag{3}$$

The transition probability of the ADHMC Markov chain augmented with MH rejection sampling using equation 3 is equal to $\mathsf{P}(Q_0, Q_K') = \int_{\mathbb{R}^d} \mathcal{P}(Q_0, Q_0', Q_K') \, \mathfrak{g}(\Pi_{Q_0}^{-f}(Q_0')) \, \mathfrak{g}(\Pi_{Q_0'}^{-b}(Q_k')) \, dQ_0'$. It is established in Ghosh et al. (2025) that the underlying Markov chain of the augmented ADHMC procedure possesses the desired time reversibility.

---

**Algorithm 1** SGHMC

Initialization: noisy gradient estimation for $\nabla U^\omega(q)$ for potential energy $U(q)$ gradient; kinetic energy $V(p) = p^2/2$ with gradient $\nabla V(p) = p$; initial iterate $q_0$; $K$ steps of size $\eta$ for total trajectory length $K\eta$

**for** $n = 1, \dots$ **do**
  Set $Q_0 \leftarrow q_{n-1}$
  Sample: $P_0 \sim \mathfrak{g}(p)$
  Lift: $(Q_0, P_0) \leftarrow Q_0$
  Move:
  Start with sample of $\nabla U^\omega(Q_0)$
  **for** $k = 0, \dots, K - 1$ **do**
    Set $P_{k+\frac{1}{2}} \leftarrow P_k - \frac{\eta}{2}\nabla U^\omega(q_k)$
    Set $Q_{k+1} \leftarrow Q_k + \eta\nabla V(P_{k+\frac{1}{2}})$
    Sample $\nabla U^\omega(Q_{k+1})$
    Set $P_{k+1} \leftarrow P_{k+\frac{1}{2}} - \frac{\eta}{2}\nabla U^\omega(Q_{k+1})$.
  **end for**
  Sample $Z \sim \text{Uniform}(0, 1)$.
  **if** $Z \leq \frac{\mathfrak{f}(Q_K)\mathfrak{g}(P_K)}{\mathfrak{f}(Q_0)\mathfrak{g}(Q_0)}$ **then**
    Project: $q_n \leftarrow (Q_K, P_K)$
  **else**
    Set $q_n \leftarrow Q_0$
  **end if**
**end for**

---

**Algorithm 2** Stochastic Gradient ADHMC

**Initialization:** noisy gradient estimation for $\nabla U^\omega(q)$ for potential energy $U(q)$ gradient; kinetic energy $V(p)$ with gradient estimation $\nabla V(p)$; initial iterate $q_0$; $K$ steps of size $\eta$ for total trajectory length $K\eta$

**for** $n = 1, \dots, N$ **do**
  Set $Q_0 \leftarrow q_{n-1}$
                      {*forward motion*}
  Sample: $P_0 \sim \mathfrak{g}(p)$
  Lift: $(Q_0, P_0) \leftarrow Q_0$
  Move:
  Start with sample of $\nabla U^\omega(Q_0)$
  **for** $k = 0, \dots, K - 1$ **do**
    Set $P_{k+\frac{1}{2}} \leftarrow P_k - \frac{\eta}{2}\nabla U^\omega(Q_k)$
    Set $Q_{k+1} \leftarrow Q_k + \eta\nabla V(P_{k+\frac{1}{2}})$
    Sample $\nabla U^\omega(Q_{k+1})$
    Set $P_{k+1} \leftarrow P_{k+\frac{1}{2}} - \frac{\eta}{2}\nabla U^\omega(Q_{k+1})$.
  **end for**
  Project: $Q_0' \leftarrow (Q_K, P_K)$
                      {*backward motion*}
  Sample: $P_0' \sim \mathfrak{g}(P)$
  Lift: $(Q_0', P_0') \leftarrow Q_0'$
  Move:
  Start with sample of $\nabla U^\omega(Q_0')$
  **for** $k = 0, \dots, K - 1$ **do**
    Set $P_{k+\frac{1}{2}}' \leftarrow P_k' + \frac{\eta}{2}\nabla U^\omega(Q_k')$
    Set $Q_{k+1}' \leftarrow q_k' - \eta\nabla V(P_{k+\frac{1}{2}}')$
    Sample $\nabla U^\omega(Q_{k+1}')$
    Set $P_{k+1}' \leftarrow P_{k+\frac{1}{2}}' + \frac{\eta}{2}\nabla U^\omega(Q_{k+1}')$.
  **end for**
  Sample $Z \sim \text{Uniform}(0, 1)$.
  **if** $Z \leq \frac{\mathfrak{f}(Q_K')\mathfrak{g}(P_K)\mathfrak{g}(P_K')}{\mathfrak{f}(Q_0)\mathfrak{g}(P_0)\mathfrak{g}(P_0')}$ **then**
    Project: $q_n \leftarrow (Q_K', Q_K')$
  **else**
    Set $q_n \leftarrow Q_0$
  **end if**
**end for**

---

# 3 Geometric Convergence of SG ADHMC

In this section, explicit geometric convergence rates are estimated for general HMC algorithms, including features such as general auxiliary distribution (ADHMC) and stochastic gradient estimation (SG). We use a *functional analysis* approach. A basic argument for geometric convergence of Markov chain, treated as iterations driven by the Markov operator, including explicit estimation of convergence rates, based on analyzing functional displacement of the operator is presented in Lovász & Simonovits (1993). The key result is that given a *time-reversible* Markov chain, with conductance $\Phi$, the functional inequality $\langle h, \mathcal{M}h \rangle \leq \left(1 - \frac{\Phi^2}{2}\right)\|h\|^2$ holds for every mean zero, non constant $h \in L^2$, where $\mathcal{M}h$ represents the image of $h$ under the Markov operator, more precisely, $\mathcal{M}h(q) = \int_{\mathbb{R}^d} h(q)P(q, dq')$. Rewriting the inequality, we have $\|h\|^2 - \langle h, \mathcal{M}h \rangle \geq \frac{\Phi^2}{2}\|h\|^2$, which says that the norm of the displacement of the Markov chain is lower

bounded by norm of the preimage up to a constant. Subsequently, the convergence rate of the Markov chain to its invariant measure in total variational distance can be quantified utilizing the above inequality, see e.g. Roberts & Rosenthal (1997), Roberts & Tweedie (2001) and Madras & Randall (2002). Recall that for two probability measures $\pi_1$ and $\pi_2$ defined on a probability triple $(\Omega, \mathcal{F}, \mathsf{P})$, their total variational distance is defined as $d_{TV}(\pi_1, \pi_2) := \sup_{A \in \mathcal{F}} |\pi_1(A) - \pi_2(A)|$. Exponential (geometric) convergence rate is defined as follow,

**Definition 4.** *For a $r \in (0, 1)$, a Markov chain is said to converge exponentially to its stationary distribution with rate at least $r$ if there exists a $C > 0$ such that for all $n \geq 1$, we have $d(\pi_n, \pi_\infty) \leq C r^n$ for certain distance between (probability) measures.*

In summary, we have the following connection between the functional inequality and the convergence rate of the Markov chain, where the convergence rate is estimated by the spectral gap of the Markov operator implied by the functional inequality.

**Lemma 3.1.** *If $\|h\|^2 - \langle h, \mathcal{M}h \rangle \geq \frac{\Phi^2}{2}\|h\|^2$ is satisfied by any mean zero $h \in L^2$, the time reversible Markov chain converges to its stationary distribution at a rate at least $1 - \frac{\Phi^2}{2}$ in total variational distance.*

We impose the following additional assumption on the curvature of $V(p)$ that will play an important role in deriving explicit expressions for the geometric convergence rates of the HMC variants.

**Assumption 4.** *Auxiliary $g(p) \propto \exp(-V(p))$ satisfies the derivative condition (with expectation under $g$):*

$$\mathbb{E}\left[\nabla V(p)\, \nabla V(p)^\top - \nabla^2 V(p)\right] = 0.$$

Assumption 4 bears a superficial resemblance to the celebrated information matrix equality (c.f. Ch 1.3.2 in Amemiya (1985)) that arises in estimation of parameters $\theta$ of a density $g_\theta(p)$ by maximizing the likelihood $L_\theta(p) = \log g_\theta(p)$. However, they are distinct because the derivatives taken there are with respect to parameters $\theta$ while the derivatives in Assumption 4 are w.r.t. $p$. In particular, the proof of the equality $\mathbb{E}\left[\nabla_\theta L_\theta(p)\nabla_\theta L_\theta(p)^\top\right] = -\mathbb{E}\left[\nabla_\theta^2 L_\theta(p)\right]$ depends on a crucial interchange between integral over $p$ and differential over $\theta$, which does not apply in the case of Assumption 4.

**Proposition 3.1.** *Suppose $\{g_m(\cdot),\ m \in [M]\}$ are a finite collection of densities, each of which satisfy Assumption 4. Let $\{\gamma_m,\ m \in [M]\}$ be a collection of real numbers such that $\gamma_m \geq 0$ and $\sum_m \gamma_m = 1$. Then, the mixture density $g(p) = \sum_m \gamma_m g_m(p)$ also satisfies Assumption 4.*

*Proof.* Let $V_m(p) = -\log g_m(p)$. The first derivative of $V(p) = -\log g(p)$ is

$$\nabla V(p) \;=\; -\frac{1}{g(p)}\sum_m \gamma_m \nabla g_m(p) \;=\; \sum_m \gamma_m \frac{g_m(p)}{g(p)}\nabla V_m(p).$$

Applying derivatives of products, we get the second derivative as

$$\nabla^2 V(p) = \sum_m \gamma_m \left[\frac{1}{g(p)}\nabla V_m(p)\nabla g_m(p)^\top \;-\; \frac{g_m(p)}{g^2(p)}\nabla V_m(p)\nabla g(p)^\top \;+\; \frac{g_m(p)}{g(p)}\nabla^2 V_m(p)\right]$$

$$= \sum_m \gamma_m \left[\frac{g_m(p)}{g(p)}\left(\nabla^2 V_m(p) - \nabla V_m(p)\nabla V_m(p)^\top\right) \;+\; \frac{g_m(p)}{g(p)}\nabla V_m(p)\nabla V(p)^\top\right]$$

$$= \sum_m \gamma_m \frac{g_m(p)}{g(p)}\left(\nabla^2 V_m(p) - \nabla V_m(p)\nabla V_m(p)^\top\right) \;+\; \left(\sum_m \gamma_m \frac{g_m(p)}{g(p)}\nabla V_m(p)\right)\nabla V(p)^\top.$$

From the expression for the first derivative $\nabla V(p)$, we see that the last term simplifies to $\nabla V(p)\nabla V(p)^\top$. The expectation w.r.t. $g(p)$ of the first term yields a zero since each density $g_m(p)$ satisfies Assumption 4. Thus, the mixture $g(p)$ also satisfies Assumption 4. $\qquad\square$

We now observe that multivariate Gaussians, the auxiliaries used in standard HMC, as well as mixtures of Gaussians, the nonsymmetric auxiliaries studied in Ghosh et al. (2025) with ADHMC, satisfy Assumption 4.

**Corollary 3.1.** *Suppose $g_m(p)$ is a multivariate Gaussian centered at $p^m$ with covariance $\Sigma_m$. Then, Assumption 4 is satisfied by $g_m(p)$. Further, mixtures $g(p) = \sum_m \gamma_m g_m(p)$ of multivariate Gaussian distributions also satisfy Assumption 4.*

*Proof.* We have, $V_m(p) = \frac{1}{2}(p - p^m)^\top \Sigma_m^{-1}(p - p^m)$ with derivatives $\nabla V_m(p) = \Sigma_m^{-1}(p - p^m)$ and $\nabla^2 V_m(p) = \Sigma^{-1}$. We get the first assertion by noting that $\mathbb{E}[(p - p^m)(p - p^m)^\top] = \Sigma_m$. The second assertion follows by applying Proposition 3.1. $\qquad\square$

The mixtures of Gaussians thus satisfy Assumption 4. They do not however satisfy Assumption 1, which is key to the analysis of geometric convergence presented in this paper. We provide a concrete example of a non-symmetric auxiliary that satisfies both Assumptions 1 and 4 in Example 1.

**Example 1.** *Consider the asymmetric auxiliary over $p \in \mathbb{R}$ with density parametrized by $\alpha \in \mathbb{R}$:*

$$g(p) \propto \begin{cases} w_1 \exp\left\{-\left(p^2 + \left(\alpha - \frac{1}{3}\right)p\right)\right\} & -\infty < p < -\alpha \\ \exp\left\{-\left(1 + \frac{1}{3\alpha}\right)\frac{p^2}{2}\right\} & -\alpha \leq p < \alpha \\ w_3 \exp\left\{-\left(\frac{p^2}{4} + \left(\frac{1}{3} + \frac{\alpha}{2}\right)p\right)\right\} & \alpha \leq p < \infty \end{cases}.$$

*It is constructed by stitching together three Gaussians on each of the three intervals. The constants $w_1$ and $w_3$ scale each segment to ensure continuity of $g(p)$ as well as its potential $V(p) = -\log g(p)$ at breakpoints $\pm\alpha$. The derivatives of $V(p)$ are*

$$V'(p) = \begin{cases} 2p + \alpha - \frac{1}{3} \\ p\left(1 + \frac{1}{3\alpha}\right) \\ \frac{p}{2} + \frac{1}{3} + \frac{\alpha}{2} \end{cases} \quad and \quad V''(p) = \begin{cases} 2 \\ 1 + \frac{1}{3\alpha} \\ \frac{1}{2} \end{cases} \quad for \quad \begin{matrix} -\infty < p < -\alpha \\ -\alpha \leq p < \alpha \\ \alpha \leq p < \infty \end{matrix}.$$

*The $V'(p)$ is continuous and together with $V''(p) > 0$ (when $\alpha > -1/3$) satisfies the conditions of Assumption 1. For $\alpha = 0.9069$, it also satisfies the conditions of Assumption 4. We present the detailed calculations in Appendix E.*

Utilizing Lemma 3.1, we have:

**Lemma 3.2.** *Under Assumptions 1 and 4, suppose that $M_H$ represents the Markov operator generated by the HMC algorithm with $K$ leapfrog steps. Then, there exist positive constants $C', C^*$ and $\eta^* = \min\{\frac{C^*\sigma_V^2}{8KA_3}, \frac{C^*}{2C'}\}$ with $C_*$ being the constant determined by Poincaré inequality for general measures and $\eta^*$ determined by $C_1, C_2,$ and $C_3$, such that when $\eta < \eta^*$, we have,*

$$\|h\|^2 - \langle h, M_H h\rangle \geq \eta^2 \left(\frac{C^*\sigma_V^2}{8} - KA_3\eta\right)\|h\|^2. \tag{4}$$

*$\sigma_V^2 := \int_{\mathbb{R}^d} \|\nabla V(p)\|_2 g(p)dp$, and the constant $A_3$ is defined in Lemma 4.3.*

The proof of Lemma 3.2 can be found in Sec. C. It leads to the following result:

**Theorem 3.1.** *Under Assumptions 1 and 4, for $\eta < \min\{\frac{C^*\sigma_V^2}{8KA_3}, \frac{C^*}{2C'}\}$ with positive constants $C'$ and $C^*$ as appeared in Lemma 3.2, the Markov chain generated by the ADHMC algorithm converges at a rate at least $1 - \eta^2\left(\frac{C^*\sigma_V^2}{8} - KA_3\eta\right)$. More precisely, we have that*

$$d_{TV}(\hat{\pi}^n, \pi) \leq \left[1 - \eta^2\left(\frac{C^*\sigma_V^2}{8} - KA_3\eta\right)\right]^n d_{TV}(\hat{\pi}, \pi),$$

*with $\hat{\pi}$ denotes the initial distribution of the Markov chain and $\hat{\pi}^n$ denotes its distribution after $n$ ADHMC transitions.*

*Proof.* The theorem follows from Lemmata 3.1 and 3.2. $\qquad\square$

In case of SGHMC, we have:

**Lemma 3.3.** *Under Assumptions 1, 3 and 4, suppose that $M_{SG}$ represents the Markov operator generated by the SGHMC algorithm with $K$ leapfrog steps. Then, for any mean zero $h \in L^2$, $\eta < \min\{\frac{C^* \sigma_V^2}{8KA_3^{SG}}, \frac{C^*}{2C'}\}$ with positive constants $C'$ and $C^*$ as appeared in Lemma 3.2, we have,*

$$\|h\|^2 - \langle h, M_{SG}h \rangle \geq \eta^2 \left( \frac{C^* \sigma_V^2}{8} - KA_3^{SG}\eta \right) \|h\|^2,$$

*where $A_3^{SG}$ is the constant from Lemma 4.4.*

*Proof.* The only difference from Lemma 3.2 are the constants estimated by Lemma 4.2. □

**Corollary 3.2.** *Under Assumptions 1, 3 and 4, for $\eta < \min\{\frac{C^* \sigma_V^2}{8KA_3^{SG}}, \frac{C^*}{2C'}\}$ with positive constants $C'$ and $C^*$ as appeared in Lemma 3.2, the Markov chain generated by the stochastic gradient ADHMC algorithm converge at a rate at least $1 - \eta^2 \left( \frac{C^* \sigma_V^2}{8} - KA_3^{SG}\eta \right)$. More precisely,*

$$d_{TV}(\hat{\pi}^n, \pi) \leq \left[ 1 - \eta^2 \left( \frac{C^* \sigma_V^2}{8} - KA_3^{SG}\eta \right) \right]^n d_{TV}(\hat{\pi}, \pi).$$

As we can see from Lemma 3.2, Theorem 3.1 and their proofs, the key for determining the convergence rates of the SGHMC algorithms lies in quantifying the closeness of the *symplectic integrator* to exact solution to the Hamiltonian system. These *quantifications* have been summarized in the next section, and detailed calculations are presented in Appendix A. From these results, we can see that the stochasticity in the gradient estimations are reflected in the calculation of the key constant $A_3^{SG}$ with the presence of the higher moments of $L^\omega$. Note that, naturally, a more volatile noisy estimation results in larger higher moments, thus enforces the step size range to be smaller, and in turn slows the convergence. Both Theorem 3.1 and Corollary 3.2 apply to ADHMC algorithms directly. For background and derivations on the *Poincaré inequality* for a general family of measures that include log-concave case, see, e.g. Mangoubi & Vishnoi (2018) and Villani (2009). For development on estimation of the best constant for the Poicaré inequality, see e.g. Cattiaux & Guillin (2020); Serres (2024). Due to this connection, as well as conditions on the moments of the auxiliary distributions, our convergence rates are less sensitive to dimension, comparing with for example those in Mangoubi & Vishnoi (2018) and Chen & Gatmiry (2023). Further precise characterization of the rate requires improved quantification of the constants involved in functional inequalities such as the Poincaré inequality, which are of wide general interest to the literature.

## 4 Technical Results and Ramifications

In this section, we present quantitative results on some key aspects of general HMC algorithms. First, we provide a range of *quantitative estimations* of numerical errors in leapfrog implementations of the Hamiltonian movement. Second, we characterize the statistical distance in *KL divergence* between two proposed HMC steps with respect to the distance between their initial states. Lastly, bounds on the (Metropolis-Hastings) *acceptance probability* are obtained. From the dependence of our main results in Section 3 on some of these results, we can see that that these quantities are crucial for the performance of SGHMC. In addition, the results presented here can also be utilized for establishing quantitative convergence results through other arguments, for example through *conductance* type arguments as presented in Chen & Gatmiry (2023).

### 4.1 Leapfrog vs Exact

One of the keys to the success of HMC algorithms is the effectiveness of the numerical symplectic integration of the Hamiltonian differential equations. Extensive efforts have been devoted to such studies for HMC Gaussian auxiliary distribution across the existing literature. Allowing *general auxiliary distributions* certainly has posed more challenges for the analysis, and the introduction of stochastic gradient estimation then requires this analysis to be combined with the probabilistic estimation of the noisy estimation. In a series of technical

results, we are able to provide sharp estimates on the error produced in these numerical procedures. Recall that the update takes the following form,

$$\hat{q} = q + \eta \nabla V \left( p - \frac{1}{2} \eta \nabla U(q) \right), \tag{5}$$

$$\hat{p} = p - \frac{1}{2} \eta \nabla U(q) - \frac{1}{2} \eta \nabla U(\hat{q}). \tag{6}$$

Meanwhile, the exact trajectory follows,

$$\dot{Q} = \nabla V(P), \quad \dot{P} = -\nabla U(Q); \qquad Q(0) = q, \quad P(0) = p.$$

Hence, we have,

$$Q(\eta) = q + \int_0^\eta \nabla V(P(t)) dt \tag{7}$$

$$P(\eta) = p - \int_0^\eta \nabla U(Q(t)) dt. \tag{8}$$

**Lemma 4.1.** *Under Assumptions 1 and 2, when* $\sup_{q \in \mathbb{R}^d} \left\| \nabla^3 U(q) \right\| < \infty$ *and* $\sup_{p \in \mathbb{R}^d} \left\| \nabla^3 V(p) \right\| < \infty$, *we have,*

$$|||Q(\eta) - \hat{q}|||_2 \quad \leq \quad \left\{ \frac{\sup_{p \in \mathbb{R}^d} \left\| \nabla^3 V(p) \right\|_{op} (d+2) L_U}{24} + \frac{L_V L_U [(d+2) L_V]^{\frac{1}{2}}}{6} \right\} \eta^3,$$

$$|||P(\eta) - \hat{p}|||_2 \quad \leq \quad (L_V(d+2))^{1/2} \times \left\{ \frac{\sup_{q \in \mathbb{R}^d} \left\| \nabla^3 U(q) \right\| (L_U)^{3/2} (d+2)^{1/2} + (L_U)^{3/2} (L_V)^{1/2}}{6} \right.$$

$$\left. + \frac{\sup_{q \in \mathbb{R}^d} \left\| \nabla^3 U(q) \right\|}{12} + \frac{(L_U)^{3/2} (L_V)^{1/2}}{4} \right\} \eta^3.$$

Lemma 4.1 is proved in Appendix as Lemmata A.3 and A.5. In case of stochastic gradient HMC, Assumptions 3 allows a similar analysis to be carried out with the presence of the stochastic gradient estimation, hence,

**Lemma 4.2.** *Under Assumptions 3, when* $\sup_{q \in \mathbb{R}^d} \left\| \nabla^3 U(q) \right\| < \infty$ *and* $\sup_{p \in \mathbb{P}} \left\| \nabla^3 V(p) \right\| < \infty$, *we have,*

$$|||Q(\eta) - \hat{q}|||_2 \quad \leq \quad \left\{ \frac{\sup_{p \in \mathbb{P}} \left\| \nabla^3 V(p) \right\|_{op} (d+2) \mathbb{E}[L_U^\omega]}{24} + \frac{L_V \mathbb{E}[L_U^\omega][(d+2) L_V]^{\frac{1}{2}}}{6} \right\} \eta^3,$$

$$|||P(\eta) - \hat{p}|||_2 \quad \leq \quad (L_V(d+2))^{1/2} \times \left\{ \frac{\sup_{q \in \mathbb{R}^d} \left\| \nabla^3 U(q) \right\| \mathbb{E}[(L_U^\omega)^{3/2}] (d+2)^{1/2} + \mathbb{E}[(L_U^\omega)^{3/2}] (L_V)^{1/2}}{6} \right.$$

$$\left. + \frac{\sup_{q \in \mathbb{R}^d} \left\| \nabla^3 U(q) \right\|}{12} + \frac{\mathbb{E}[(L_U^\omega)^{3/2}] (L_V)^{1/2}}{4} \right\} \eta^3.$$

## 4.2 Continuity of the Step in Probability Space

A key observation for HMC is that for a pair of starting points $q_1$ and $q_2$, the distance of probability measures, as measured for instance by the Kullback-Leibler (KL) divergence, for the next step of the algorithm is

bounded by the linear order of the distance between $q_1$ and $q_2$. Therefore, when these two points are close, the next step distributions are also similar. Hence, it is easy to see that this is a key step in a conductance based argument for geometric convergence, as seen in Chen & Gatmiry (2023).

For a fixed $q \in \mathbb{R}^d$, the probability measure $\mathcal{P}_q$ of the image $Q \in \mathbb{R}^d$ can be viewed as a pushforward of the auxiliary probability measure via the integrator. Its density is given by the expression (9), where $p(q, Q)$ denotes the inverse of the integrator (as $\Pi_Q^{-f}(q)$ in equation 3):

$$\mathfrak{g}(p(q, Q)) \det\left(\frac{\partial p(q, Q)}{\partial Q}\right) . \tag{9}$$

For any pair $q_1, q_2 \in \mathbb{R}^d$, the Kullback-Leibler(KL) divergence $KL(\mathcal{P}_{q_1} || \mathcal{P}_{q_2})$ can be written as,

$$KL(\mathcal{P}_{q_1} || \mathcal{P}_{q_2}) = \int_{\mathbb{R}^d} \left\{ \log \mathfrak{g}(p) - \log[\mathfrak{g}(p(q_2, Q(q_1, p)))] - \log \det\left(\frac{\partial p(q_2, Q(q_1, p))}{\partial Q}\right) \right\} \mathfrak{g}(p) dp.$$

Propositions 4.1 and 4.2, which are not directly employed in obtaining the results of this paper, generalize some key elements of various other approaches of convergence analysis for HMC and its variants, such as the *conductance analysis* in Chen & Gatmiry (2023) or the *probabilistic analysis* in Mangoubi & Vishnoi (2018). Hence, they are be utilized in the analysis of related algorithms for auxiliary functions considered in this paper.

**Proposition 4.1.** *For HMC with general auxiliary distribution, we have,*

$$KL(\mathcal{P}_{q_1} || \mathcal{P}_{q_2}) \leq \frac{\eta \|\nabla^3 V\| L_U}{2} \|q_1 - q_2\|.$$

In addition, we have the similar result for stochastic gradient estimate,

**Proposition 4.2.** *For SGHMC, we have*

$$KL(\mathcal{P}_{q_1} || \mathcal{P}_{q_2}) \leq \frac{\eta \|\nabla^3 V\| \mathbb{E}[L_U^\omega]}{2} \|q_1 - q_2\|.$$

The proof of Propositions 4.1 and 4.2 can be found in Sec.D.

### 4.3 Lower bounding the Acceptance Probability

Another key component in all convergence analysis of HMC with numerical integrator is the estimation (bounding) of the acceptance probability of the proposed motion. This is eventually reduced to the estimation of the deviation of the numerical integrator from the exact solution in terms of the function value. More specifically, we have,

**Lemma 4.3.** *Under Assumptions 1 and 2, we have $\mathbb{E}|U(\hat{q}) - U(Q(\eta))| + \mathbb{E}|V(\hat{p}) - V(P(\eta))| \leq K A_3 \eta^3$, with*

$$A_3 := \left[L_U \|\|q\|\|_2 + (dL_U)^{1/2}\right] \left\{ \frac{(d+2)T_V L_U}{24} + \frac{L_V L_U [(d+2)L_V]^{\frac{1}{2}}}{6} \right\} +$$

$$+ \left[L_V \|\|p\|\|_2 + (dL_U)^{1/2}\right] (L_V(d+2))^{1/2} \left\{ \frac{T_U (L_U)^{3/2}(d+2)^{1/2} + (L_U)^{3/2}(L_V)^{1/2}}{6} + \right.$$

$$\left. + \frac{T_U}{12} + \frac{(L_U)^{3/2}(L_V)^{1/2}}{4} \right\} .$$

Similarly, for SGHMC, we have,

**Lemma 4.4.** *Under Assumptions 1 and 2 for V and Assumptions 3 for the stochastic implementations, we have* $\mathbb{E}|U(\hat{q}) - U(Q(\eta))| + \mathbb{E}|V(\hat{p}) - V(P(\eta))| \le K A_3^{SG} \eta^3$, *with*

$$A_3^{SG} := \left\{ \frac{(d+2)T_V[\mathbb{E}[(L_U^\omega)^2] + d^{1/2}\mathbb{E}[L_U^\omega]^{3/2}]}{24} + \frac{L_V[\mathbb{E}[(L_U^\omega)^2] + d^{1/2}\mathbb{E}[(L_U^\omega)^{3/2}][(d+2)L_V]^{\frac{1}{2}}}{6} \right\}$$

$$+ L_V |||p|||_2 (L_V(d+2))^{1/2} \left\{ \frac{\bar{T}\mathbb{E}[L_U^\omega)^{3/2}]](d+2)^{1/2} + \mathbb{E}[L_U^\omega)^{3/2}](L_V)^{1/2}}{6} \right.$$

$$+ \frac{\bar{T}}{12} + \frac{\mathbb{E}[L_U^\omega)^{3/2}](L_V)^{1/2}}{4} \Bigg\}$$

$$+ d^{1/2} \left\{ \frac{\bar{T}\mathbb{E}[(L_U^\omega)^2](d+2)^{1/2} + \mathbb{E}[(L_U^\omega)^2](L_V)^{1/2}}{6} + \frac{\bar{T}\mathbb{E}[(L_U^\omega)^{1/2}]}{12} + \frac{\mathbb{E}[(L_U^\omega)^2](L_V)^{1/2}}{4} \right\}.$$

The proofs of these lemmas can be found in Sec.B. This naturally leads to the following result.

**Proposition 4.3.** *Under Assumptions 1 and 2 ( Assumptions 1 and 2 for V and Assumptions 3 for SGHMC): For any* $\varrho, \delta \in (0, 1)$, *one can choose* $K$ *and* $\eta$ *such that for subset* $D \subseteq \mathbb{R}^d \times \mathbb{R}^d$ *and* $\mathsf{P}[(q, p) \in D] \ge 1 - \delta$, *the acceptance probability is lower bounded by* $\varrho$.

## 5 Conclusions

In this paper, we analyzed the convergence of the HMC method where (a) the gradient is accessed only via a noisy estimator (that is, as a Stochastic Gradient), (b) Hamiltonian motion is discretized with leapfrog steps and (c) the auxiliary takes a general nonsymmetric form that requires HMC to be modified to take steps with alternating directions. We take an analytic method approach to our investigation and derive bounds on the geometric convergence rates for a large family of SGHMC algorithms with general auxiliary distributions.

As more applications of HMC emerge from different areas of machine learning, we expect these results to allow the presented algorithms to be adapted more readily and with higher confidence. We also expect the analytic methods developed in the paper can be more extensively utilized in the analysis of algorithms in this domain.

For future research, we would like to complete an analysis of ADHMC convergence by relaxing the Lipschitz-smoothness condition on $\nabla V$ in Assumption 1 to allow for mixtures of Gaussians. Another goal is to explore HMC algorithms on Riemannian manifolds and path spaces. For HMC on a Riemannian manifold, see e.g. Girolami & Calderhead (2011). In these methods, the form of auxiliary distribution takes the state variable $q$ as a parameter, and this poses a new challenge to its convergence analysis. For HMC on path spaces such as those proposed in Pinski (2021), these are defined on infinite dimensional spaces, which requires new methodologies for its understanding. We expect that the Dirichlet form based method developed in this paper to be key elements for quantitative analysis of these algorithms.

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

**Appendix**

In the appendix, we present some fundamental estimations that are both crucial for establishing the main results in the paper and of independent interests. First, in Sec. A, we present a detailed estimation between the exact Hamiltonian calculation and the leapfrog integrator; then, in Sec. B, we provide a lower bound to the acceptance probability. Then, the proof of the key Lemma 3.2 is presented in C. In Sec. D we quantify the dependence of the leapfrog implementations upon the initial state in terms of divergence of probability measure.

## A    Error Estimation for Leapfrog Gradient Calculations

In this section, we quantify the errors between the leapfrog implementation, including the case with stochastic gradient, and the exact Hamiltonian solutions. These results will be used later in multiple occasions, such as the estimation of the lower bound on acceptance probabilities and eventual convergence rates. We will start with a lemma which is utilized in later derivation.

**Lemma A.1.** *Under Assumption 1, we have, for any integer $\ell \geq 1$,*

$$|||\nabla U(q)|||^2_\ell = \left[ \mathbb{E}_{q \sim \exp(-U)} \|\nabla U(q)\|^{2\ell}_2 \right]^{\frac{1}{\ell}} \leq (d + 2\ell - 2)L_U, \tag{10}$$

$$|||\nabla V(p)|||^2_\ell = \left[ \mathbb{E}_{p \sim \exp(-V)} \|\nabla V(p)\|^{2\ell}_2 \right]^{\frac{1}{\ell}} \leq (d + 2\ell - 2)L_V. \tag{11}$$

*Proof.* Assumption 1 implies that $Tr(\nabla^2 U) \leq L_U d$, inequalities (10) and (11) follow from Lemma 9 of Chen & Gatmiry (2023), note that the subexponential condition is naturally satisfied. □

### A.1    Difference between $\hat{q}$ and $Q(\eta)$

We have:

$$\hat{q} - Q(\eta) = \int_0^\eta \left[ \nabla V \left( p - \frac{1}{2}\eta \nabla U(q) \right) - \nabla V(P(t)) \right] dt$$

$$\text{follows from equations equation 5 and  equation 7}$$

$$= \int_0^\eta \int_0^1 \nabla^2 V \left( s \left( p - \frac{1}{2}\eta \nabla U(q) \right) + (1-s)P(t) \right) \left[ \left( p - \frac{1}{2}\eta \nabla U(q) \right) - P(t) \right] ds\, dt$$

$$\text{an application of Newton-Leibniz}$$

$$= \int_0^\eta \int_0^1 \nabla^2 V \left( s \left( p - \frac{1}{2}\eta \nabla U(q) \right) + (1-s)P(t) \right) \left( \int_0^t \nabla U(Q(\tau))d\tau - \frac{1}{2}\eta \nabla U(q) \right) ds\, dt,$$

$$\text{follows from equations equation 6 and  equation 8}$$

$$= \underbrace{\int_0^\eta \int_0^1 \nabla^2 V \left( s \left( p - \frac{1}{2}\eta \nabla U(q) \right) + (1-s)P(t) \right) \left( \int_0^t [\nabla U(Q(\tau)) - \nabla U(q)]d\tau \right) ds\, dt}_{A_1} +$$

$$+ \underbrace{\int_0^\eta \int_0^1 \nabla^2 V \left( s \left( p - \frac{1}{2}\eta \nabla U(q) \right) + (1-s)P(t) \right) \left( t - \frac{1}{2}\eta \right) \nabla U(q)\, ds\, dt}_{A_2}$$

$$\text{write the term } \tfrac{1}{2}\eta \nabla U(q) \text{ as } \tfrac{1}{2}\eta \nabla U(q) - \int_0^t \nabla U(q)d\tau + t\nabla U(q).$$

Now, examine $A_2$

$$A_2 = \int_0^\eta \int_0^1 \nabla^2 V \left( s \left( p - \frac{1}{2}\eta\nabla U(q) \right) + (1-s)P(t) \right) \left( t - \frac{1}{2}\eta \right) \nabla U(q) \, ds \, dt,$$

$$= \int_0^1 \int_0^\eta \nabla^2 V \left( s \left( p - \frac{1}{2}\eta\nabla U(q) \right) + (1-s)P(t) \right) \left( t - \frac{1}{2}\eta \right) \nabla U(q) \, dt \, ds$$

exchange the order of integrations

$$= \int_0^1 \int_0^\eta \left[ \nabla^2 V \left( s \left( p - \frac{1}{2}\eta\nabla U(q) \right) + (1-s)P(t) \right) - \nabla^2 V \left( s \left( p - \frac{1}{2}\eta\nabla U(q) \right) + (1-s)p \right) \right] \cdot$$

$$\cdot \left( t - \frac{1}{2}\eta \right) \nabla U(q) \, dt \, ds +$$

$$+ \int_0^1 \int_0^\eta \nabla^2 V \left( s \left( p - \frac{1}{2}\eta\nabla U(q) \right) + (1-s)p \right) \left( t - \frac{1}{2}\eta \right) \nabla U(q) \, dt \, ds,$$

$$= \int_0^1 \int_0^\eta \left[ \nabla^2 V \left( s \left( p - \frac{1}{2}\eta\nabla U(q) \right) + (1-s)P(t) \right) - \nabla^2 V \left( s \left( p - \frac{1}{2}\eta\nabla U(q) \right) + (1-s)p \right) \right] \cdot$$

$$\cdot \left( t - \frac{1}{2}\eta \right) \nabla U(q) \, dt \, ds$$

because $\nabla^2 V \left( s \left( p - \frac{1}{2}\eta\nabla U(q) \right) + (1-s)p \right)$ is independent of $t$ and $\int_0^\eta (t - \frac{1}{2}\eta) dt = 0$

$$= \int_0^1 \int_0^\eta \int_0^t \nabla^3 V \left( s \left( p - \frac{1}{2}\eta\nabla U(q) \right) + (1-s)P(\tau) \right) \nabla U(Q(\tau)) \left( t - \frac{1}{2}\eta \right) \nabla U(q) \, d\tau \, dt \, ds$$

an application of Newton-Leibniz

$$= \int_0^1 \int_0^\eta \int_\tau^\eta \nabla^3 V \left( s \left( p - \frac{1}{2}\eta\nabla U(q) \right) + (1-s)P(\tau) \right) \nabla U(Q(\tau)) \left( t - \frac{1}{2}\eta \right) \nabla U(q) \, dt \, d\tau \, ds$$

exchange the order of integrations with respect to $t$ and $\tau$

$$= \int_0^1 \int_0^\eta \nabla^3 V \left( s \left( p - \frac{1}{2}\eta\nabla U(q) \right) + (1-s)P(\tau) \right) \nabla U(Q(\tau)) \frac{\tau}{2}(\eta - \tau)\nabla U(q) \, d\tau \, ds$$

integrate out $t$

The above calculations can be summarized as

**Lemma A.2.** *The difference between the leapfrog update with step $\eta$ and the trajectory at time $\eta$ is equal to:*

$$\hat{q} - Q(\eta)$$
$$= \int_0^\eta \int_0^1 \nabla^2 V \left( s \left( p - \frac{1}{2}\eta\nabla U(q) \right) + (1-s)P(t) \right) \left( \int_0^t \nabla[U(Q(\tau)) - \nabla U(q)]d\tau \right) \, ds \, dt$$
$$+ \int_0^1 \int_0^\eta \nabla^3 V \left( s \left( p - \frac{1}{2}\eta\nabla U(q) \right) + (1-s)P(\tau) \right) \nabla U(Q(\tau)) \frac{\tau}{2}(\eta - \tau)\nabla U(q) \, d\tau \, ds \qquad (12)$$

This will for the basic for quantifying $|||Q(\eta) - \hat{q}|||_2$.

**Lemma A.3.** *Under Assumptions 1 and 2, we have,*

$$|||Q(\eta) - \hat{q}|||_2 \leq \left\{ \frac{(d+2)L_U T_V}{24} + \frac{L_V L_U [(d+2)L_V]^{\frac{1}{2}}}{6} \right\} \eta^3, \qquad (13)$$

*Proof.* Let us look at the two terms in equation 12.

$$\left\|\left\|\left\| \int_0^\eta \int_0^1 \nabla^2 V \left( s \left( p - \frac{1}{2}\eta\nabla U(q) \right) + (1-s)P(t) \right) \left( \int_0^t \nabla[U(Q(\tau)) - \nabla U(q)]d\tau \right) \, ds \, dt \right\|\right\|\right\|_2$$

$$=\left\|\left\|\left\| \int_0^\eta \int_0^1 \nabla^2 V \left( s \left( p - \frac{1}{2}\eta\nabla U(q) \right) + (1-s)P(t) \right) \left( \int_0^t \int_0^\tau \nabla^2 U(Q(w))\nabla V(P(w))dw; d\tau \right) \, ds \, dt \right\|\right\|\right\|_2$$

an application of Newton-Leibniz

$$\leq \int_0^\eta \int_0^1 \int_0^t \int_0^\tau \left\|\left\|\left\| \nabla^2 V \left( s \left( p - \frac{1}{2}\eta\nabla U(q) \right) + (1-s)P(t) \right) \nabla^2 V(P(w))\nabla V(P(w)) \right\|\right\|\right\|_2 dw; d\tau \, ds \, dt$$

$$\leq \int_0^\eta \int_0^1 \int_0^t \int_0^\tau L_U L_V \left\|\left\|\left\| \nabla V(P(w)) \right\|\right\|\right\|_2 dw \, d\tau \, ds \, dt$$

due to assumption 1

$$\leq L_V L_U [(d+2)L_V]^{\frac{1}{2}}/6.$$

due to LemmaA.1 for $p = 2$ and the fact that the joint probability is invariant under the Hamilton motion

$$\left\|\left\|\left\| \int_0^1 \int_0^\eta \nabla^3 V \left( s \left( p - \frac{1}{2}\eta\nabla U(q) \right) + (1-s)P(\tau) \right) \nabla U(Q(\tau))\frac{\tau}{2}(\eta - \tau)\nabla U(q) \, d\tau \, ds \right\|\right\|\right\|_2$$

$$\leq \int_0^1 \int_0^\eta \|\nabla^3 V\|\frac{\tau}{2}(\eta - \tau)\mathbb{E}\left[ \left\|\left\| \nabla U(Q(\tau)) \right\|\right\|_2 \cdot \left\|\left\| \nabla U(q) \right\|\right\|_2 \right] d\tau \, ds$$

assumption on the operator norm of the third degree tensor

$$\leq \int_0^1 \int_0^\eta \|\nabla^3 V\|\frac{\tau}{2}(\eta - \tau)\left\|\left\| \nabla U(Q(\tau)) \right\|\right\|_2 \cdot \left\|\left\| \nabla U(q) \right\|\right\|_2 d\tau \, ds$$

Hölder's inequality

$$\leq \|\nabla^3 V\|L_U(d+2)/24.$$

$\square$

## A.2 Difference between $\hat{p}$ and $P(\eta)$

Again, let us start with a decomposition of the term $P(\eta) - \hat{p}$,

$$\hat{p} - P(\eta) = \int_0^\eta \nabla U(Q(t)) - \frac{1}{2}\nabla U(q) - \frac{1}{2}\nabla U(\hat{q}) \, dt$$

follows from equations equation 6 and equation 8

$$= \int_0^\eta [\nabla U(Q(t)) - \nabla U(q)] \, dt - \frac{1}{2}\int_0^\eta [\nabla U(q) - \nabla U(\hat{q})] \, dt$$

$$= \int_0^\eta \int_0^t \nabla^2 U(Q(s))\nabla V(P(s)) \, ds \, dt - \frac{1}{2}\int_0^\eta [\nabla U(q) - \nabla U\left( q + \eta\nabla V\left( p - \frac{1}{2}\eta\nabla U(q) \right) \right)] \, dt$$

an application of Newton-Leibniz

$$= \underbrace{\int_0^\eta \int_0^t \nabla^2 U(Q(s))\nabla V(P(s)) \, ds \, dt - \frac{1}{2}\int_0^\eta [\nabla U(q) - \nabla U(q + \eta\nabla V(p))] \, dt}_{B_1}$$

$$+ \underbrace{\frac{1}{2}\int_0^\eta \left[ \nabla U(q + \eta\nabla V(p)) - \nabla U\left( q + \eta\nabla V\left( p - \frac{1}{2}\eta\nabla U(q) \right) \right) \right] \, dt}_{B_2}.$$

$$B_2 = \frac{1}{2} \int_0^\eta \left[ \nabla U(q + \eta \nabla V(p)) - \nabla U \left( q + \eta \nabla V \left( p - \frac{1}{2} \eta \nabla U(q) \right) \right) \right] dt$$
$$= \frac{1}{2} \int_0^\eta \int_0^\eta \left[ \nabla^2 U \left( q + s \nabla V(p) + (1-s) \nabla V \left( p - \frac{1}{2} \eta \nabla U(q) \right) \right) \left[ \nabla V(p) - \nabla V \left( p - \frac{1}{2} \eta \nabla U(q) \right) \right] \right] ds \, dt.$$

an application of Newton-Leibniz

$B_2$ is clearly a $\eta^3$ term.

$$B_1 = \int_0^\eta \int_0^t \nabla^2 U(Q(s)) \nabla V(P(s)) \, ds \, dt - \frac{1}{2} \int_0^\eta [\nabla U(q) - \nabla U(q + \eta \nabla V(p))] \, dt$$
$$= \int_0^\eta \int_0^t \nabla^2 U(Q(s)) \nabla V(P(s)) \, ds \, dt - \frac{1}{2} \int_0^\eta \int_0^\eta [\nabla^2 U(q + s \nabla V(p))] \nabla V(p) \, dt$$

an application of Newton-Leibniz

$$= \underbrace{\int_0^\eta \int_0^t \nabla^2 U(Q(s)) \nabla V(P(s)) \, ds \, dt - \int_0^\eta \int_0^t [\nabla^2 U(q + s \nabla V(p))] \nabla V(P(s)) \, ds \, dt}_{B_{11}}$$
$$+ \underbrace{\int_0^\eta \int_0^t [\nabla^2 U(q + s \nabla V(p))] \nabla V(P(s)) \, ds \, dt - \frac{1}{2} \int_0^\eta \int_0^\eta [\nabla^2 U(q + s \nabla V(p))] \nabla V(p) \, dt}_{B_{12}}.$$

For $B_{11}$, we have,

$$B_{11} = \int_0^\eta \int_0^t \nabla^2 U(Q(s)) \nabla V(P(s)) \, ds \, dt - \int_0^\eta \int_0^t [\nabla^2 U(q + s \nabla V(p))] \nabla V(P(s)) \, ds \, dt$$
$$= \int_0^\eta \int_0^t [\nabla^2 U(Q(s)) - \nabla^2 U(q + s \nabla V(p))] \nabla V(P(s)) \, ds \, dt$$

$$B_{12} = \int_0^\eta \int_0^t [\nabla^2 U(q + s \nabla V(p))] \nabla V(P(s)) \, ds \, dt - \frac{1}{2} \int_0^\eta \int_0^\eta [\nabla^2 U(q + s \nabla V(p))] \nabla V(p) \, dt$$
$$= \underbrace{\int_0^\eta \int_0^t [\nabla^2 U(q + s \nabla V(p))] \nabla V(P(s)) \, ds \, dt - \int_0^\eta \int_0^t [\nabla^2 U(q + s \nabla V(p))] \nabla V(p) \, ds \, dt}_{B_{121}}$$
$$+ \underbrace{\int_0^\eta \int_0^t [\nabla^2 U(q + s \nabla V(p))] \nabla V(p) \, ds \, dt - \frac{1}{2} \int_0^\eta \int_0^\eta [\nabla^2 U(q + s \nabla V(p))] \nabla V(p) \, dt}_{B_{122}}.$$

Hence, we have the following expression,

**Lemma A.4.**

$$P(\eta) - \hat{p}$$
$$= \int_0^\eta \int_0^t [\nabla^2 \nabla^2 U(Q(s)) - U(q + s \nabla V(p))] \nabla V(P(s)) \, ds \, dt$$
$$+ \int_0^\eta \int_0^t [\nabla^2 U(q + s \nabla V(p))] \nabla V(P(s)) \, ds \, dt - \int_0^\eta \int_0^t [\nabla^2 U(q + s \nabla V(p))] \nabla V(p) \, ds \, dt$$
$$+ \int_0^\eta \int_0^t [\nabla^2 U(q + s \nabla V(p))] \nabla V(p) \, ds \, dt - \frac{1}{2} \int_0^\eta \int_0^\eta [\nabla^2 U(q + s \nabla V(p))] \nabla V(p) \, dt$$
$$+ \frac{1}{2} \int_0^\eta \int_0^\eta \left[ \nabla^2 U \left( q + s \nabla V(p) + (1-s) \nabla V \left( p - \frac{1}{2} \eta \nabla U(q) \right) \right) \left[ \nabla V(p) - \nabla V \left( p - \frac{1}{2} \eta \nabla U(q) \right) \right] \right] ds \, dt.$$

**Lemma A.5.** *Under Assumptions 1 and 2, we have,*

$$|||P(\eta) - \hat{p}|||_2$$
$$\leq (L_V(d+2))^{1/2} \left\{ \frac{T_U(L_U)^{3/2}(d+2)^{1/2} + (L_U)^{3/2}(L_V)^{1/2}}{6} + \frac{T_U}{12} + \frac{(L_U)^{3/2}(L_V)^{1/2}}{4} \right\} \eta^3. \tag{14}$$

*Proof.* From above derivations, we know that, $|||P(\eta) - \hat{p}|||_2 \leq |||B_{11}|||_2 + |||B_{121}|||_2 + |||B_{122}|||_2 + |||B_2|||_2$. Therefore,

$$|||B_{11}|||_2 \leq \int_0^\eta \int_0^t |||\nabla^2 U(Q(s))\nabla V(P(s)) - [\nabla^2 U(q + s\nabla V(p))]\nabla V(P(s))|||_2 \, ds \, dt$$

$$\leq \sup_{q \in \mathbb{R}^d} \left\| \nabla^3 U(q) \right\| \int_0^\eta \int_0^t \int_0^s \mathbb{E} \left[ \left\| \nabla V(P(\tau)) - \nabla V(p) \right\|_2 \cdot \left\| \nabla V(P(s)) \right\|_2 \right] ds \, dt$$

     assumption on the operator norm of the third degree tensor

$$\leq \sup_{q \in \mathbb{R}^d} \left\| \nabla^3 U(q) \right\| \int_0^\eta \int_0^t \int_0^s \left\| \left\| \nabla V(P(\tau)) - \nabla V(p) \right\| \right\|_2 \cdot \left\| \left\| \nabla V(P(s)) \right\| \right\|_2 ds \, dt$$

     Cauchy-Schwarz inequality

$$\leq \sup_{q \in \mathbb{R}^d} \left\| \nabla^3 U(q) \right\| L_V [L_V(d+2)]^{1/2} \int_0^\eta \int_0^t \int_0^s \left\| \left\| P(\tau) - p \right\| \right\|_2 ds \, dt$$

     Lemma A.1 and Lipschitz condition of $\nabla V$

$$\leq \frac{\sup_{q \in \mathbb{R}^d} \left\| \nabla^3 U(q) \right\| (L_U)^{1/2}(L_V)^{3/2}(d+2)}{6} \eta^3.$$

     Lemma A.1

$$|||B_{121}|||_2 \leq \int_0^\eta \int_0^t \left\| \left\| [\nabla^2 U(q + s\nabla V(p))]\nabla V(P(s)) - \int_0^t [\nabla^2 U(q + s\nabla V(p))]\nabla V(p) \right\| \right\|_2 ds \, dt$$

$$\leq L_U \int_0^\eta \int_0^t \left\| \left\| \nabla V(P(s)) - \nabla V(p) \right\| \right\|_2 ds \, dt$$

     assumption on $U$

$$\leq L_U L_V \int_0^\eta \int_0^t \left\| \left\| P(s) - p \right\| \right\|_2 ds \, dt$$

     assumption on $V$

$$\leq L_U L_V \int_0^\eta \int_0^t \int_0^s |||\nabla U(\tau)|||_2 \, d\tau \, ds \, dt$$

     an application of Newton-Leibniz

$$\leq \frac{(L_U)^{3/2} L_V (d+2)^{1/2}}{6} \eta^3.$$

Apply Lemma A.6, we have,

$$|||B_{122}|||_2 \leq \left\| \left\| \int_0^\eta \int_0^t [\nabla^2 U(q + s\nabla V(p))]\nabla V(p) \, ds \, dt - \frac{1}{2} \int_0^\eta \int_0^\eta [\nabla^2 U(q + s\nabla V(p))]\nabla V(p) \, dt \right\| \right\|_2$$

$$\leq \frac{\sup_{q \in \mathbb{R}^d} \left\| \nabla^3 U(q) \right\| (L_V)^{1/2}(d+2)^{1/2}}{12} \eta^3.$$

$$||| B_2 |||_2 \leq \frac{1}{2} \int_0^\eta \int_0^\eta \left\| \left\| \nabla^2 U \left( q + s\nabla V(p) + (1-s)\nabla V \left( p - \frac{1}{2}\eta\nabla U(q) \right) \right) \left[ \nabla V(p) - \nabla V \left( p - \frac{1}{2}\eta\nabla U(q) \right) \right] \right\| \right\|_2 ds \, dt$$

$$\leq \frac{1}{2} \int_0^\eta \int_0^\eta L_U \left\| \left\| \left[ \nabla V(p) - \nabla V \left( p - \frac{1}{2}\eta\nabla U(q) \right) \right] \right\| \right\|_2 ds \, dt$$

$$\leq \frac{(L_U)^{3/2} L_V (d+2)^{1/2}}{4} \eta^3.$$

□

The following technical lemma and its proof are included for completeness.

**Lemma A.6.** *For a locally integrable function $f(\cdot)$, we have,*

$$\int_0^\eta \int_0^t f(s) \, ds \, dt - \frac{1}{2} \int_0^\eta \int_0^\eta f(s) \, ds \, dt = \int_0^\eta \frac{\tau}{2}(\tau - \eta) f'(\tau) \, d\tau.$$

*Proof.*

$$\int_0^\eta \int_0^t f(s) \, ds \, dt - \frac{1}{2} \int_0^\eta \int_0^\eta f(s) \, ds \, dt$$

$$= \int_0^\eta \int_s^\eta f(s) \, dt \, ds - \frac{\eta}{2} \int_0^\eta f(s) \, ds$$

$$= \int_0^\eta f(s)(\eta - s) \, ds - \frac{\eta}{2} \int_0^\eta f(s) \, ds$$

$$= \int_0^\eta f(s)(\frac{\eta}{2} - s) \, ds$$

$$= \int_0^\eta [f(s) - f(0)](\frac{\eta}{2} - s) \, ds$$

$$= \int_0^\eta \int_0^s f'(\tau) \, d\tau (\frac{\eta}{2} - s) \, ds$$

$$= \int_0^\eta \int_\tau^\eta f'(\tau)(\frac{\eta}{2} - s) \, ds \, d\tau$$

$$= \int_0^\eta \frac{\tau}{2}(\tau - \eta) f'(\tau) \, d\tau.$$

□

# B  Lower Bounding the Acceptance Probability

From the expression of the acceptance/rejection probability calculation in equation 2, we know that it suffices to show that there exists $a > 0$, such that, $|U(q_K) + V(p_K) - U(q_0) - V(p_0)| \leq a$ since $\{(q_K, p_K) : \exp[U(q_K) + V(p_K) - U(q_0) - V(p_0)] < e^a\} = \{(q_K, p_K) : U(q_K) + V(p_K) - U(q_0) - V(p_0) < a\}$. Meanwhile, since the Hamiltonian is invariant for the exact solution, this quantity become the difference between the exact Hamiltonian and that of the symplectic integrator. In essence, we need to estimate $|U(\hat{q}) - U(Q(\eta))|$ and $|V(\hat{p}) - V(P(\eta))|$. Lemmata A.3 and A.5 imply that these terms are of order $\eta^3$. Then the desired results follows from applying them $K$ times. This is a similar result to those in Chen & Gatmiry (2023) where it is stated, for general subexponential target probability distribution, there exists a compact set $\Lambda \in \mathbb{R}^d \times \mathbb{R}^d$, such that when the initial point $(q_0, p_0) \in \Lambda$, the acceptance can be bounded from below.

$$\mathbb{E}|U(\hat{q}) - U(Q(\eta))| \leq \mathbb{E}\left|\int_0^1 \nabla U(q_s) \cdot [\hat{q} - Q(\eta)]ds\right|$$

$$\leq \mathbb{E}\left|\int_0^1 [\nabla U(q_s) - \nabla U(q)] \cdot [\hat{q} - Q(\eta)]ds\right| + \int_0^1 \mathbb{E}\left|\nabla U(q) \cdot [\hat{q} - Q(\eta)]ds\right|$$

$$\leq L_U \mathbb{E}\left|\int_0^1 \int_0^s [q_\tau - q] \cdot [\hat{q} - Q(\eta)] \, d\tau \, ds\right| + \int_0^1 \mathbb{E}\left|\nabla U(q) \cdot [\hat{q} - Q(\eta)]ds\right|$$

$$\leq L_U \int_0^1 \int_0^s |||q_\tau - q|||_2 |||[\hat{q} - Q(\eta)]|||_2 \, d\tau \, ds + |||\nabla U(q)|||_2 |||[\hat{q} - Q(\eta)]|||_2$$

$$\leq (L_U |||q|||_2 + |||\nabla U(q)|||_2) |||\hat{q} - Q(\eta)|||_2$$

$$\leq [L_U |||q|||_2 + (dL_U)^{1/2}] \left\{ \frac{T_V(d+2)L_U}{24} + \frac{L_V L_U [(d+2)L_V]^{\frac{1}{2}}}{6} \right\} \eta^3,$$

with $q_s = (s\hat{q} + (1-s)Q(\eta))$. All the quantities are available through Lemmata A.1 and A.3. By the same arguments, we can obtain the following estimate for $\mathbb{E}|V(\hat{p}) - V(P(\eta))|$ which can be bounded further with Lemmata A.1 and A.5

$$\mathbb{E}|V(\hat{p}) - V(P(\eta))|$$

$$\leq (L_V |||p|||_2 + |||\nabla V(q)|||_2)|||) |||\hat{p} - P(\eta)|||_2$$

$$\leq [L_V |||p|||_2 + (dL_U)^{1/2}](L_V(d+2))^{1/2} \left\{ \frac{T_U(L_U)^{3/2}(d+2)^{1/2} + (L_U)^{3/2}(L_V)^{1/2}}{6} \right.$$

$$\left. + \frac{T_U}{12} + \frac{(L_U)^{3/2}(L_V)^{1/2}}{4} \right\} \eta^3.$$

## C  Proof of Lemma 3.2

*Proof of Lemma 3.2.* We will first show that inequality (4) holds for Schwarz functions $h$ from $\mathbb{R}^d$ to $\mathfrak{f}(q)dq$ with uniform constants, then a standard mollification procedure, see e.g. Lieb & Loss (2001), leads the inequality for all $h \in L^2(\mathbb{R}^d, \mathfrak{f}(q)dq)$. Moreover, Theorem 2.16 in Lieb & Loss (2001) and its remarks assure that for there exist constants $C$ and $C'$ we can have $\|\nabla h\|_\infty \leq C\|h\|_2, \|D^3 h\|_\infty \leq C'\|h\|_2$ and $C \leq \frac{C^*}{8K}$ which is the optimal Poincaré constant for general distributions by carefully choosing the convolution functions.

First, consider the case $K = 1$, we have,

$$\int_{\mathbb{R}^d} h(q) \int_{\mathbb{R}^d} [h(q) - h(\hat{q})]g(p)dpdq$$

$$= \int_{\mathbb{R}^d} h(q) \int_{\mathbb{R}^d} \left[h(q) - h(q + \eta \nabla V(p - \frac{\eta}{2}\nabla U(q)))\right] g(p)dpdq$$

$$= \underbrace{\int_{\mathbb{R}^d} h(q) \int_{\mathbb{R}^d} [h(q) - h(q + \eta \nabla V(p))]g(p)dpdq}_{III_1}$$

$$+ \underbrace{\int_{\mathbb{R}^d} h(q) \int_{\mathbb{R}^d} \left[h(q + \eta \nabla V(p)) - h(q + \eta \nabla V(p - \frac{\eta}{2}\nabla U(q)))\right] g(p)dpdq}_{III_2}.$$

Then, for $III_1$, we have, from mean value theorem with $\tilde{q}$ satisfying $\|\tilde{q} - q\|_\infty \leq \|\nabla V(p)\|_\infty$ such that $h(q + \eta \nabla V(p)) = h(q) + \eta \nabla h \cdot \nabla V(p) + \frac{1}{2}\nabla V(\tilde{q})^\top \nabla^2 h(q)\nabla V(\tilde{q})$.

$$III_1 = \int_{\mathbb{R}^d} h(q) \int_{\mathbb{R}^d} [h(q) - h(q + \eta \nabla V(p))]g(p)dpdq$$

$$= \underbrace{-\frac{\eta^2}{2} \int_{\mathbb{R}^d} \nabla V(p) \cdot \left[ \int_{\mathbb{R}^d} h(q)\nabla^2 h(q)f(q)dq \right] \cdot \nabla V(p)g(p)dp}_{III_{11}}$$

$$+ \underbrace{\frac{\eta^2}{2} \int_{\mathbb{R}^d} \nabla V(p) \cdot \left[ \int_{\mathbb{R}^d} h(q)[\nabla^2 h(q) - \nabla^2 h(\tilde{q})]f(q)dq \right] \cdot \nabla V(p)g(p)dp,}_{III_{12}}$$

due to the zero mean assumption of $p$ and mean value theorem with $\tilde{q}$.

Define the mean joint partial derivatives $\mu_{ij} = \int_{\mathbb{R}^d} \frac{\partial V(p)}{\partial p_i} \frac{\partial V(p)}{\partial p_j} \mathfrak{g}(p)dp$ and the mean second derivative $\sigma_{ij} = \int_{\mathbb{R}^d} \frac{\partial^2 V(p)}{\partial p_i \partial p_j} \mathfrak{g}(p)dp$. Note that Assumption 4 gives that $\mu_{ij} = \sigma_{ij}$ for all $i, j$.

Then, for $III_{11}$, apply integration by part, we have,

$$III_{11} = -\frac{\eta^2}{2} \int_{\mathbb{R}^d} \nabla V(p) \cdot \left[ \int_{\mathbb{R}^d} h(q)\nabla^2 h(q)f(q)dq \right] \cdot \nabla V(p)g(p)dp$$

$$= \frac{\eta^2}{2} \int_{\mathbb{R}^d} \nabla V(p) \cdot \int_{\mathbb{R}^d} [\nabla h(q) \otimes \nabla h(q)f(q) + h(q)\nabla h(q) \otimes \nabla f(q)]dq \cdot \nabla V(p)g(p)dp$$

$$= \frac{\eta^2 \sigma_V^2}{2} \int_{\mathbb{R}^d} \left[ \|\nabla h(q)\|_2^2 - \sum_{i,j} h(q)\partial_i h(q)\partial_j U(q) \right] f(q)dq \cdot \mu_{ij}.$$

with $\sigma_V^2 := \int_{\mathbb{R}^d} \|\nabla V(p)\|_2 g(p)dp$. For $III_{12}$, for any $i, j$

$$\left| \int_{\mathbb{R}^d} h(q)D_{ij}h(q) - D_{ij}h(\tilde{q})]f(q)dq \right| \le \|h\|_2^2 \|\|D_{ij}h(q) - D_{ij}h(\tilde{q})\|\| \le C'\eta \|h\|_2^2$$

where the first inequality is due to the Hölder's inequality and the second inequality is based on the assumption on the third derivatives as we explained above. Next,

$$III_2 = \int_{\mathbb{R}^d} h(q) \int_{\mathbb{R}^d} \left[ h(q + \eta \nabla V(p)) - h(q + \eta \nabla V(p - \frac{\eta}{2}\nabla U(q))) \right] g(p)dpdq$$

$$= \int_{\mathbb{R}^d} h(q)\nabla h(q) \cdot \nabla U(q)]f(q)dq \cdot \int_{\mathbb{R}^d} \nabla^2 V(p)g(p)dp,$$

by the convexity assumption on $U(q)$. Meanwhile, we have,

$$\int_{\mathbb{R}^d} h(q)[\nabla h(q) \cdot \nabla U(q)]f(q)dq \cdot \int_{\mathbb{R}^d} \nabla^2 V(p)g(p)dp, = \sum_{i,j} \int_{\mathbb{R}^d} h(q)[h(q)\partial_i h(q)\partial_j U(q)]f(q)dq \cdot \sigma_{ij}.$$

Therefore, the cancellation follows from the assumption $\sigma_{ij} = \mu_{ij}$. Putting everything together, we have,

$$\int_{\mathbb{R}^d} h(q) \int_{\mathbb{R}^d} [h(q) - h(\hat{q})]g(p) = \frac{\eta^2 \sigma_V^2}{2} \|\nabla h(q)\|_2^2 + III_{12}$$

Since we know that $III_{12} \le C_2 \eta^3 \|h\|_2$. For $\eta < \frac{C^*}{2C_2}$, with $C^*$ being the optimal Poincaré inequality constant, we have, $\int_{\mathbb{R}^d} h(q) \int_{\mathbb{R}^d} [h(q) - h(\hat{q})]g(p)dpdq \ge \frac{\varrho \eta^2}{4}C^* \sigma_V^2 \|h\|_2$. For general $K > 1$, from the fact that $p$ has zero mean, we have,

$$\int_{\mathbb{R}^d} \int_{\mathbb{R}^d} [h(q_k) - h(q_{k+1})]^2 g(p)dpdq \le C\eta^2 \sigma_V^2 \|h\|_2^2 \le \frac{C^*}{8K} \sigma_V^2 \|h\|_2^2,$$

where the second inequality follows from the relationship between $C$ and $C^*$. Thus,

$$\int_{\mathbb{R}^d} h(q) \int_{\mathbb{R}^d} [h(q) - h(q_K)] g(p) dp dq$$

$$= \sum_{k=0}^{K-2} [h(q_k) - h(q_{k+1})]^2 g(p) dp dq + \int_{\mathbb{R}^d} h(q_{K-1}) \int_{\mathbb{R}^d} [h(q_{K-1}) - h(q_K)] g(p) dp dq$$

$$\geq \frac{\varrho \eta^2}{8} C^* \sigma_V^2 \|h\|_2.$$

applying the inequality for $K = 1$ to the last term and the previous inequality with $q_0 = q$.

Denote $A(q, p)$ as the event that the proposal is accepted. $\|h\|_2^2 - \langle h, M_H h \rangle = \int_{\mathbb{R}^d} h(q) \int_{\mathbb{R}^d} [h(q) - h(\hat{q})] f(q) g(p) dp dq + \int_{\mathbb{R}^d} h(q) \int_{\mathbb{R}^d} [1 - \mathbf{1}_{A(q,p)}][h(q) - h(\hat{q})] g(p) dp dq$. We can bound the second term with higher order of $K A_3 \eta^3 \|h\|_2^2$ (at least $\eta^3$) using Hölder's inequality and Lemma 4.3. Therefore, for $\eta < \eta^* = \min\{\frac{C^* \sigma_V^2}{8 K A_3}, \frac{C^*}{2C'}\}$, we have, $\|h\|_2^2 - \langle h, M_H h \rangle \geq \eta^2 (\frac{C^* \sigma_V^2}{8} - K A_3 \eta) \|h\|_2^2$. $\qquad \square$

# D   Density of Pushforward Auxiliary Distributions

Fix $q \in \mathbb{R}^d$, the probability measure $\mathcal{P}_q$ of the image $Q \in \mathbb{R}^d$ can be viewed as a pushforward of the auxiliary probability measure via the integrator, therefore, its density bears the following form,

$$\mathfrak{g}(p(q, Q)) \det\left(\frac{\partial p(q, Q)}{\partial Q}\right), \tag{15}$$

with $p(q, Q)$ denotes the inverse of the integrator.

**Kullback-Leibler(KL) divergence calculation**

For any pair $q_1, q_2 \in \mathbb{R}^d$, the Kullback-Leibler(KL) divergence $KL(\mathcal{P}_{q_1} \| \mathcal{P}_{q_2})$ can be written as,

$$KL(\mathcal{P}_{q_1} \| \mathcal{P}_{q_2}) = \int_{\mathbb{R}^d} \mathfrak{g}(p(q_1, Q)) \det\left(\frac{\partial p(q_1, Q)}{\partial Q}\right) \log\left(\frac{\mathfrak{g}(p(q_1, Q)) \det\left(\frac{\partial p(q_1, Q)}{\partial Q}\right)}{\mathfrak{g}(p(q_2, Q)) \det\left(\frac{\partial p(q_2, Q)}{\partial Q}\right)}\right) dQ$$

$$\overset{(1)}{=} \int_{\mathbb{R}^d} \log\left(\frac{\mathfrak{g}(p) \det\left(\frac{\partial p(q_1, Q(q_1, p))}{\partial Q}\right)}{\mathfrak{g}(p(q_2, Q(q_1, p))) \det\left(\frac{\partial p(q_2, Q(q_1, p))}{\partial Q}\right)}\right) \mathfrak{g}(p) dp$$

$$\overset{(2)}{=} \int_{\mathbb{R}^d} \log\left(\frac{\mathfrak{g}(p)}{\mathfrak{g}(p(q_2, Q(q_1, p))) \det\left(\frac{\partial p(q_2, Q(q_1, p))}{\partial Q}\right)}\right) \mathfrak{g}(p) dp$$

$$= \int_{\mathbb{R}^d} \left(\log \mathfrak{g}(p) - \log[\mathfrak{g}(p(q_2, Q(q_1, p)))] - \log \det\left(\frac{\partial p(q_2, Q(q_1, p))}{\partial Q}\right)\right) \mathfrak{g}(p) dp,$$

equation (1) is the result of change of variable from $Q$ to $p$, and (2) is due to the fact that,

$$\frac{\partial p(q_1, Q(q_1, p))}{\partial Q} = Id.$$

Note that, conceptually, the term $p(q_2, Q(q_1, p))$ is treated as perturbation of $p$, denoted as $\tilde{p} = p + \epsilon$, then it can be see that

$$\int_{\mathbb{R}^d} \left(\log \mathfrak{g}(p) - \log[\mathfrak{g}(p(q_2, Q(q_1, p)))] - \log \det\left(\frac{\partial p(q_2, Q(q_1, p))}{\partial Q}\right)\right) \mathfrak{g}(p) dp$$

$$= \int_{\mathbb{R}^d} \left(\log\left(\frac{\mathfrak{g}(p)}{\mathfrak{g}(\tilde{p})}\right) - \log \det \partial \mathfrak{g}(\tilde{p})\right) \mathfrak{g}(p) dp$$

$$= \int_{\mathbb{R}^d} \left(\log(1 + \epsilon_1) - \log \det(I + \epsilon_2)\right) \mathfrak{g}(p) dp.$$

Now, let us examine them more carefully. Recall that $\mathfrak{g}(p) = \exp[-V(p)]$, so,

$$\log\left(\frac{\mathfrak{g}(p)}{\mathfrak{g}(\tilde{p})}\right) = V(\tilde{p}) - V(p). \tag{16}$$

Let us see how $\tilde{p}$ is calculated. We start with the case that only one leapfrog step is taken,

$$\hat{q}_1 = q_1 + \eta\nabla V\left(p - \frac{1}{2}\eta\nabla U(q_1)\right),$$

$$\hat{p}_1 = p - \frac{1}{2}\eta\nabla U(q_1) - \frac{1}{2}\eta\nabla U(\hat{q}_1).$$

So $\tilde{p}$ satisfies,

$$q_2 + \eta\nabla V\left(\tilde{p} - \frac{1}{2}\eta\nabla U(q_2)\right) = q_1 + \eta\nabla V\left(p - \frac{1}{2}\eta\nabla U(q_1)\right).$$

Therefore, we can compute $\frac{\partial\tilde{p}}{\partial p}$,

$$\eta\nabla^2 V\left(\tilde{p} - \frac{1}{2}\eta\nabla U(q_2)\right)\frac{\partial\tilde{p}}{\partial p} = \eta\nabla^2 V\left(p - \frac{1}{2}\eta\nabla U(q_1)\right),$$

Hence,

$$\frac{\partial\tilde{p}}{\partial p} = \left(\nabla^2 V\left(\tilde{p} - \frac{1}{2}\eta\nabla U(q_2)\right)\right)^{-1}\nabla^2 V\left(p - \frac{1}{2}\eta\nabla U(q_1)\right).$$

Furthermore, we know that,

$$\frac{\partial p(q_1, Q)}{\partial Q} = \frac{\partial\tilde{p}}{\partial p}\cdot\frac{\partial p}{\partial Q}.$$

These calculations will make an estimation of equation 16 possible. More specifically, Hessian Lipschitz condition will lead to

$$\frac{\partial\tilde{p}}{\partial p} = \left(\nabla^2 V\left(\tilde{p} - \frac{1}{2}\eta\nabla U(q_2)\right)\right)^{-1}\nabla^2 V\left(p - \frac{1}{2}\eta\nabla U(q_1)\right)$$

$$= I - \left(\nabla^2 V\left(\tilde{p} - \frac{1}{2}\eta\nabla U(q_2)\right)\right)^{-1}\left[\nabla^2 V\left(\tilde{p} - \frac{1}{2}\eta\nabla U(q_2)\right) - \nabla^2 V\left(p - \frac{1}{2}\eta\nabla U(q_1)\right)\right].$$

We know that

$$\left(\nabla^2 V\left(\tilde{p} - \frac{1}{2}\eta\nabla U(q_2)\right)\right)^{-1}$$

is bounded ($\succeq mI$ by strong convexity). Therefore, we only need to bound

$$\left\|\nabla^2 V\left(\tilde{p} - \frac{1}{2}\eta\nabla U(q_2)\right) - \nabla^2 V\left(p - \frac{1}{2}\eta\nabla U(q_1)\right)\right\|.$$

Hessian Lipschitz condition leads to

$$\left\|\nabla^2 V\left(\tilde{p} - \frac{1}{2}\eta\nabla U(q_2)\right) - \nabla^2 V\left(p - \frac{1}{2}\eta\nabla U(q_1)\right)\right\|$$

$$\leq \frac{\|\nabla^3 V\|\eta}{2}\|\nabla U(q_1) - \nabla U(q_2)\| \leq \frac{\eta\|\nabla^3 V\|L_U}{2}\|q_1 - q_2\|. \tag{17}$$

Inequality equation 17 thus give Propositions 4.1 and 4.2, similar to Lemma 2 in Chen & Gatmiry (2023).

## E   Detailed Calculations for Example 1.

The Example 1 is a special case of a four-parameter $(\alpha, \delta, \gamma_1$ and $\gamma_2)$ class of distributions with piecewise linear derivatives for the potential $V(p)$ :

$$
V'(p) = \begin{cases} (1+\gamma_1)p + \alpha\gamma_1 - \delta \\ \left(1+\frac{\delta}{\alpha}\right)p \\ (1+\gamma_2)p + \delta - \gamma_2\alpha \end{cases} \quad \text{and} \quad V''(p) = \begin{cases} 1+\gamma_1 \\ 1+\frac{\delta}{\alpha} \\ 1+\gamma_2 \end{cases} \quad \text{for} \quad \begin{array}{c} -\infty < p < -\alpha \\ -\alpha \le p < \alpha \\ \alpha \le p < \infty \end{array}.
$$

With the limits that $\gamma_1, \gamma_2 > -1$ and $\alpha > -\delta$, these kinetic energy functions satisfy the conditions of Assumption 1. The corresponding density $g(p)$ is :

$$
g(p) \propto \begin{cases} w_1 \exp\left\{-\left((1+\gamma_1)\frac{p^2}{2} + (\alpha\gamma_1 - \delta)\,p\right)\right\} & -\infty < p < -\alpha \\ 1 \times \exp\left\{-\left(1+\frac{\delta}{\alpha}\right)\frac{p^2}{2}\right\} & -\alpha \le p < \alpha \\ w_3 \exp\left\{-\left((1+\gamma_2)\frac{p^2}{2} + (\delta - \alpha\gamma_2)\,p\right)\right\} & \alpha \le p < \infty \end{cases}.
$$

Scaling values $w_1$ and $w_3$ are chosen to ensure that the density expressions on either side of the breakpoints $\pm\alpha$ agree in value.

We check the conditions of Assumption 4 for the asymmetric distribution. The condition requires that

$$
I \quad := \quad \int_{-\infty}^{\infty} \left[(V'(p))^2 - V''(p)\right] e^{-V(p)} dp = 0.
$$

We will break $I$ into three summands along the three intervals over which $g(p)$ is defined. We recall that for a standard Gaussian with density $\phi(u) = e^{-u^2/2}/\sqrt{2\pi}$ and cumulative distribution function $\Phi(u)$, we have the following indefinite integral forms:

$$
\Phi(b) - \Phi(a) := \int_a^b \phi(u)du; \quad \text{and} \quad \int_a^b u^2\phi(u)du = \left[\Phi(u) - u\phi(u)\right]_a^b.
$$

An implication that we will use: $\int_a^b (u^2 - 1)\phi(u)du = \left[-u\phi(u)\right]_a^b$.

For the left semi-infinite interval we have:

$$
I_1 = w_1 \int_{-\infty}^{-\alpha} \left(((1+\gamma_1)p + \alpha\gamma_1 - \delta)^2 - (1+\gamma_1)\right) \exp\left\{-\left((1+\gamma_1)\frac{p^2}{2} + (\alpha\gamma_1 - \delta)\,p\right)\right\} dp
$$

$$
\text{substituting} \quad z = p\sqrt{1+\gamma_1} + \frac{\alpha\gamma_1 - \delta}{\sqrt{1+\gamma_1}} \quad \text{with range} \quad \left(-\infty, \ \bar{z}_1 = \frac{-(\alpha+\delta)}{\sqrt{1+\gamma_1}}\right) \tag{18}
$$

$$
= w_1 \sqrt{2\pi(1+\gamma_1)} \ \exp\left\{\frac{1}{2}\left(\frac{\alpha\gamma_1 - \delta}{\sqrt{1+\gamma_1}}\right)^2\right\} \int_{-\infty}^{\bar{z}_1} (z^2 - 1)\, e^{-\frac{z^2}{2}}\, \frac{1}{\sqrt{2\pi}}dz
$$

$$
\text{from indefinite integral for} \quad \phi(\cdot)
$$

$$
= w_1 \sqrt{2\pi(1+\gamma_1)} \ \exp\left\{\frac{1}{2}\left(\frac{\alpha\gamma_1 - \delta}{\sqrt{1+\gamma_1}}\right)^2\right\} \left[-z\phi(z)\right]_{-\infty}^{\bar{z}_1}
$$

$$
\text{and symmetry of} \quad \phi(\cdot)
$$

$$
= w_1 \,(\alpha+\delta)\, \sqrt{2\pi} \ \exp\left\{\frac{1}{2}\left(\frac{\alpha\gamma_1 - \delta}{\sqrt{1+\gamma_1}}\right)^2\right\} \, \phi\left(\frac{(\alpha+\delta)}{\sqrt{1+\gamma_1}}\right).
$$

The right side semi-infinite interval similarly uses the substitution

$$z = p\sqrt{1+\gamma_2} + \frac{\delta - \alpha\gamma_2}{\sqrt{1+\gamma_2}} \quad \text{with range} \quad \left[\frac{\alpha+\delta}{\sqrt{1+\gamma_2}}, \ \infty\right) \tag{19}$$

to yield

$$I_3 = w_3 \int_\alpha^\infty \left( ((1+\gamma_2)p + \delta - \alpha\gamma_2)^2 - (1+\gamma_2) \right) \ \exp\left\{ - \left( (1+\gamma_2)\frac{p^2}{2} + (\delta - \alpha\gamma_2)\,p \right) \right\} \ dp$$

$$= \ w_3 \ (\alpha+\delta) \ \sqrt{2\pi} \ \exp\left\{ \frac{1}{2}\left(\frac{\delta - \alpha\gamma_2}{\sqrt{1+\gamma_2}}\right)^2 \right\} \ \phi\left(\frac{(\alpha+\delta)}{\sqrt{1+\gamma_2}}\right).$$

For the integral over the bounded interval in the middle:

$$I_2 = w_3 \int_{-\alpha}^\alpha \left( \left(1+\frac{\delta}{\alpha}\right)^2 p^2 - \left(1+\frac{\delta}{\alpha}\right) \right) \ \exp\left\{ - \left(1+\frac{\delta}{\alpha}\right)\frac{p^2}{2} \right\} \ dp$$

substituting $z = p\sqrt{1+\frac{\delta}{\alpha}}$ with range $[-\bar{z}_2, \ \bar{z}_2)$ where $\bar{z}_2 := \alpha\sqrt{1+\frac{\delta}{\alpha}}$ $\tag{20}$

$$= w_3\sqrt{2\pi\left(1+\frac{\delta}{\alpha}\right)} \ \left[- z\phi(z)\right]_{-\bar{z}_2}^{\bar{z}_2} \ = \ -2w_3\alpha\sqrt{2\pi}\left(1+\frac{\delta}{\alpha}\right) \ \phi\left(\alpha\sqrt{1+\frac{\delta}{\alpha}}\right).$$

Putting the three together as $I(\alpha,\delta,\gamma_1,\gamma_2) = I_1 + I_2 + I_3$ gives us a four-parameter function for the integral for which we need to seek a zero in order to satisfy Assumption 4. The specific Example 1 was constructed by fixing $\delta = 1/3$, $\gamma_1 = 1.0$ and $\gamma_2 = -1/2$ and numerically searching for a zero of $I(\alpha, 1/3, 1., -1/2)$ over $\alpha > -1/3$ to obtain $\alpha = 0.9069$.

Samples can be generated from stitched Gaussians of form equation **??** by noting the variable transformations equation 18, equation 20 and equation 19 respectively for the Gaussians in the three intervals to the standard Gaussian distribution. To start with, we first need to calculate the relative probabilities $v_i$ under the Gaussians in each interval. Samples can be generated as follows:

1. Generate $U \sim \text{Uniform}(0,1)$.

2. If $U \le \frac{v_1}{v_1+v_2+v_3}$, return a sample from $\mathcal{N}\left(-\frac{\alpha\gamma_1-\delta}{1+\gamma_1}, \ \frac{1}{\sqrt{1+\gamma_1}}\right)$ from within $(-\infty, -\alpha)$.

3. Else if $U \le \frac{v_1+v_2}{v_1+v_2+v_3}$, return a sample from $\mathcal{N}\left(0, \ \frac{1}{\sqrt{1+\frac{\delta}{\alpha}}}\right)$ from within $[-\alpha, \alpha)$.

4. Else return a sample from $\mathcal{N}\left(\frac{\alpha\gamma_2-\delta}{1+\gamma_2}, \ \frac{1}{\sqrt{1+\gamma_2}}\right)$ from within $[\alpha, -\infty)$.

