# OpenReview forum: "On Convergence of the Alternating Directions Stochastic Gradient Hamiltonian Monte Carlo (SGHMC) Algorithms"
_TMLR — Decision pending for TMLR_

### Review · Reviewer_WjEg · 2026-04-26

**Summary Of Contributions:**

This paper provides a theoretical study of HMC algorithms with an auxiliary distribution not necessarily symmetric, and the gradient of the potential approximated. The authors show that the resulting Markov chain converges towards their stationary distribution and give their convergence rate. These results hold for potential and kinetic energies being strongly convex and smooth.

While the results of the paper seem valid, I found that the paper is not clearly written, and there are a lot of tools which are not introduced, or only briefly. The paper is not well organized, which makes it pretty hard to read. Also, I am not sure of the validity of all the claims.

**Strengths**:
- Convergence results for HMC with potentially asymmetric auxiliary distribution and approximated gradient of the potential

**Weaknesses**:
- Paper not well written, hard to follow, not well organized
- Some claims are not very clear
- I have some doubts on some claims (e.g. that the convergence rate holds for the Mixture of Gaussian auxiliary distribution)
- No experiments to verify the claims

**Audience:**

Yes

**Audience Explanation:**

MCMC algorithms are of interest to the Machine Learning community. Hence, the results of this paper might interest a part of TMLR's audience.

**Claims And Evidence:**

No

**Claims Explanation:**

I did not read all the proofs as I am not an expert of HMC's algorithm. However, I have some concerns and questions about some results, which I will detail in the requested changes part.

**Requested Changes:**

In definition 2, for which $t$ is there $\phi_\epsilon(t)\in [-\epsilon, 1+\epsilon]$? Also, I think it would be better to write, "there exists a real function $\phi_\epsilon:\mathbb{R}\to\mathbb{R}$".

For the Dirichlet form described page 5, what would be $\phi_\epsilon$?

Why is the spectral gap introduced in Section 2? It does not seem to be used again.

I think the preliminary results of Section 2.4 could go to Appendix as no context nor any explanation is given. Moreover, Lemma 2.1 is only used in Appendix and Lemma 2.2 does not seem to be used?

Before Lemma 3.1, it is written "More specifically," and then the lemma is given. You should end the sentence...

In Lemma 3.1, the convergence rate is in Total Variation, which is not recalled. Definition 4 should be before Lemma 3.1 as it recalls the definition of exponential convergence of a Markov chain.

In Theorem 3.1, the convergence rate is given for the Markov chain generated by the HMC algorithm. Does it correspond to ADHMC? Same question for Corollary 2. If it is well ADHMC, then the text of the theorem and corollary are misleading in my opinion.

The result of Theorem 3.1 is said to be valid under Assumption 1 and 4. While Assumption 4 is satisfied by Mixture of Gaussians, I don't think Assumption 1 is as the resulting potential is not strongly convex? If I am not mistaken here, this would mean that this does not hold for Mixture of Gaussian?

In some of the Lemmas, there are constants determined by Poincaré's inequality. With strongly log-concave distributions, don't we have access to the constant?

Are the results of Section 4.1 used in the convergence results of Section 3?



**Typos**:
- No year in the citation of "Kaibo Wang, Jinguang Chen, Huali Yang, Han Sun, et al. Hamiltonian monte carlo based neural process for few-shot knowledge graph completion. Jinguang and Yang, Huali and Sun, Han, Hamiltonian Monte Carlo Based Neural Process for Few-Shot Knowledge Graph Completion."
- The following reference is twice in the bibliography: "Soumyadip Ghosh, Yingdong Lu, and Tomasz Nowicki. Hamiltonian monte carlo with asymmetrical momentum distributions. Physica D: Nonlinear Phenomena, 483:134952, 2025b. ISSN 0167-2789. doi: https://doi.org/10.1016/j.physd.2025.134952. URL https://www.sciencedirect.com/science/article/pii/S0167278925004294."
- Page 2: "tail.Their": lack space
- In Assumption 3: "Leap-Frog"
- Before Equation (2), I think $(Q_K,Q_K)$ and $(Q_0,Q_0)$ should be respectively $(Q_K,P_K)$ and $(Q_0,P_0)$?
- Page 6: "Thus, if the Hamiltonian motion implementation were exact, there will be no sample rejection." -> there would be
- In the algorithms, the line Lift: $(Q_0,P_0)<-Q_0$ feels weird.
- Before Lemma 3.1, it is written $Mf$ instead of $\mathcal{M}f$. There is also two times $dq'$ in the integral.
- First paragraph Section 4: "In addition, The"

---

> ### Author Response · Authors · 2026-04-29
> **Responses to Reviewer WjEg**
>
> We thank the reviewer WjEg for the careful reading and insightful comments on our work. We provide detailed
> responses to each of the requested changes. We will also provide an updated article that reflects all the
> changes we describe here and corrects all the typos once we have had a chance to respond to the feedback from all the reviewers. Please bear with us till all the reviews are be posted.
>
>
> *Qn: In definition 2, for which $t$ is there $\phi_\epsilon\in[-\epsilon, 1+\epsilon]$? Also, I think it would be better to write, "there exists a real function $\phi_\epsilon:{\mathbb R} \to {\mathbb R}$".*
>
> **Our response: Thank you for pointing out this vagueness.  $\phi_\epsilon\in[-\epsilon, 1+\epsilon]$ is the condition for all $t\in {\mathbb R}$, we have modified the definition to be in the following form.**
>
> **Definition 2. A symmetric bilinear form ${\mathcal E}(\cdot, \cdot)$ with domain $D[{\mathcal E}]$ on the Hilbert space $L^2(X,m)$ with $X$ being a metric space and $m$ a Borel measure is Markovian if for any $\epsilon>0$, there exists a real function $\phi_\epsilon(t):{\mathbb R}\to {\mathbb R}$ satisfying**
>
> **1.$ \phi_\epsilon(t) \in [-\epsilon, 1+\epsilon]$, $\forall t\in{\mathbb R}$,**
>
> **2.$\phi_\epsilon(t) = t$ for $t\in[0,1]$, and**
>
> **3.$0\le \phi_\epsilon(t')-\phi_\epsilon(t)\le t'-t$ whenever $t<t'$,**
>
> **such that $u\in D[{\mathcal E}]$ implies that $\phi_\epsilon(u)\in D[{\mathcal E}]$ and ${\mathcal E}(\phi_\epsilon(u), \phi_\epsilon(u))\le {\mathcal E}(u,u)$.**
>
> *Qn: For the Dirichlet form described page 5, what would be $\phi_\epsilon$?*
>
> **Our response: For the Dirichlet form described in page 5 the $\phi_\epsilon(t)$ can be chosen to be a mollifier that is typical in analysis. The following example is given in Example & Exercise 1.2.1 in the book by Fukushima, Oshima, and Takeda that is list in References:**
> $\phi_\epsilon(t) = (j_\delta* \psi_\epsilon)(t)$,
> **a convolution of two functions with $\psi_\epsilon(x) :=\min [(1+\epsilon), \max(-\epsilon, x)]$, and $j_\delta (x)=\delta^{-1}j(\delta^{-1}x)$ for some $\delta \in (0, \epsilon)$, where $j(x):=\gamma \exp(-1/(1-|x|^2))\chi_{(-1,1)}(x)$ with $\chi_A$ being the characteristic function of set $A$.**
>
> *Qn:Why is the spectral gap introduced in Section 2? It does not seem to be used again.*
>
> **Our response:  The spectral gap is closely related to the convergence rate of the Markov chain, and it is the key idea behind Lemma 3.1 which is a well-established result we quoted. We will rewrite the relevant sections to clarify this connection.**
>
> *Qn: I think the preliminary results of Section 2.4 could go to Appendix as no context nor any explanation is given. Moreover, Lemma 2.1 is only used in Appendix and Lemma 2.2 does not seem to be used?*
>
> **Our response: Thank you for the suggestion. We will move these materials to appendix. Indeed the original role of Lemma 2.2 was replaced by some improved technical estimates, therefore we no longer use it and will remove it.**
>
> *Qn: Before Lemma 3.1, it is written "More specifically," and then the lemma is given. You should end the sentence...*
>
> **Our response: We are happy to do that change.**

---

> ### Author Response · Authors · 2026-04-29
> **Responses to Reviewer WjEg, cont.**
>
> *Qn: In Lemma 3.1, the convergence rate is in Total Variation, which is not recalled. Definition 4 should be before Lemma 3.1 as it recalls the definition of exponential convergence of a Markov chain.*
>
> **Our response: Thank you for pointing this out. We will move Definition 4 before Lemma 3.1, and recall the definition of total variation distance right after it.**
>
> *Qn: In Theorem 3.1, the convergence rate is given for the Markov chain generated by the HMC algorithm. Does it correspond to ADHMC? Same question for Corollary 2. If it is well ADHMC, then the text of the theorem and corollary are misleading in my opinion.*
>
> **Our response: Thank you for pointing this out. We will make more explicit in each result we present the exact algorithm that it pertains to. Theorem 3.1 analyzes the ADHMC with exact gradient calculations.**
>
> *Qn: The result of Theorem 3.1 is said to be valid under Assumption 1 and 4. While Assumption 4 is satisfied by Mixture of Gaussians, I don't think Assumption 1 is as the resulting potential is not strongly convex? If I am not mistaken here, this would mean that this does not hold for Mixture of Gaussian?*
>
> **Our response: Mixture-of-Gaussians (MoG) is a class of distributions that satisfy Assumption 4, and it is provided here to stress that Assumption 4 is not very restrictive. We appreciate your point that MoG does not satisfy strong convexity everywhere as required by Assumption 1. For a treatment of ADHMC with exact gradient calculations for the class of MoG, please see the following paper that is also cited in reference.**
>
> **S. Ghosh, Y. Lu, and T. Nowicki. Hamiltonian Monte Carlo with asymmetrical momentum distributions. Physica D: Nonlinear Phenomena, 483:134952, 2025a.**
>
> *Qn: In some of the Lemmas, there are constants determined by Poincar\'e's inequality. With strongly log-concave distributions, don't we have access to the constant?*
>
> **Our response: Finding best constant for Poincar\'e's inequality is an active research area. We would like to provide the following references, and will add them to the paper.**
>
>
> **Cattiaux, P., Guillin, A. (2020). On the Poincaré Constant of Log-Concave Measures. In: Klartag, B., Milman, E. (eds) Geometric Aspects of Functional Analysis. Lecture Notes in Mathematics, vol 2256. Springer, Cham.**
>
> **Serres, J. (2024). Behavior of the Poincar\'e constant along the Polchinski renormalization flow. Communications in Contemporary Mathematics, 26(07), 2350035.**
>
> *Qn: Are the results of Section 4.1 used in the convergence results of Section 3?*
>
> **Our response:  Thank you for pointing this out. The two inequalities in Lemma 4.1 are re-stated and proved as Lemma A.2 and Lemma A.4 in the Appendix. They are the key estimations in the proof of the critical result Lemma 3.2, and thus referenced several times in its proof in the form of the inequalities given by Lemma A.2 and A.4. Lemma 3.1 in turn leads to Theorem 3.1. Similarly, their counterparts in the case of stochastic gradient estimation are the two inequalities in Lemma 4.2 that play the same role for Lemma 3.3 and Corollary 2. Thank you for bringing to our attention this double-statement of the same results, which led to the confusion that you encountered. We will iron this out and ensure that the proofs in the Appendix are clearly seen to be those of Lemma 4.1 and 4.2, and that their importance to Lemma 3.2 and 3.3 are transparent.**

---

> > ### Comment · Reviewer_WjEg · 2026-06-24
> >
> > Thank you for your response.
> >
> > I am still concerned, as I don't understand which auxiliary distributions besides the Gaussian satisfy the result of the paper? I believe you should at least provide few examples. As for now, it seems the only other distribution mentioned is the Mixture of Gaussian which does not satisfy Assumption 1...

---

### Review · Reviewer_xgZe · 2026-05-31

**Summary Of Contributions:**

The paper establishes convergence rates for HMC sampling algorithms, where target density gradients are estimated by mini-batching and Hamiltonian dynamics are approximated via leapfrog integration. A connection of these rates to the quality of the leapfrog implementation is shown. Importantly, the authors present results for general auxiliary distributions. The key tool used to obtain the theoretical results are the Dirichlet forms associated with the Markov chain induced by the sampling algorithm.


Hamiltonian Monte Carlo is often considered the gold standard for Bayesian inference of high-dimension posteriors, for instance in Bayesian deep learning. Establishing theoretical guarantees for these algorithms is of great use to the community.

While the theoretical results are very interesting, their presentation would benefit from some more refinement (see below).

**Audience:**

Yes

**Audience Explanation:**

HMC methods are often considered the gold standard for sampling high dimensional probability distributions. A better theoretical understanding of these algorithms is certainly of interest to the research community.

**Broader Impact Concerns:**

None.

**Claims And Evidence:**

Yes

**Claims Explanation:**

All theoretical statements are accompanied by corresponding proofs or references, which, to the best of my knowledge, support the respective claims.

**Requested Changes:**

### Questions

* Section 1, "Celebrated result of Jeff Cheeger", what is this, can you cite it?

* Section 1.1, “Adapting the kinetic energy V(p) to have similar multi-modal characteristics to the potential U(q) helps accelerate HMC identify its minima”; of course we want the auxiliary to be as close to the target as possible, but how do we achieve this? Even if we allow the auxiliary to be multimodal, where should these modes be? It seems like quite advanced knowledge of the target is necessary for this. And could we not make things worse than with a wide uni-modal auxiliary, if we put the modes in unfortunate locations?

* Section 2.3.1, "Stochastic oracle" sounds like an oxymoron, is this the established way to refer to stochastic gradient estimates in the HMC literature?

* Section 1.1, what is meant by “global optimal minima of U(q)”?

* For the results in Section 3, do we know something about C$^*$?

* Lemma 3.2, since they are not in Eq. (6), where do C$_1$ and C$_2$ actually enter?

* Section 3: “It is certainly desirable to obtain more precise results on the rate”; what do you mean by this? The final sentence of this section could be sharpened.

* You say Lemmas 4.3 and 4.4 are not used in your results, but are key elements of other published work. Are these lemmas then not already given in these previous works? In general, why state them in the main body if they, as you say, do not enter into your own contributions?

### Comments

* Section 1, “In many practical situations, the gradient of the potential energy U(q) of the target density function, which is essential for the running of HMC, is not available or difficult to compute.” At least one example would be nice here.

* Section 1, the final paragraph of the introduction discusses advanced technical concepts at such a high level that there is little benefit to the reader. It is good to introduce the main technique used in this work in the introduction, but technical details should be left for the main body.

* Definition 3, “The Dirichlet form approach on the convergence rate is closely related the study of conductance originated in Cheeger (1969) and carried out by a series of subsequent studies.” and also following sentence, should be corrected in terms of grammar.

* Section, 2.3.2, “ξ is a random variable for uniformly randomly pick a size-B ”, should be corrected for grammar.

* Why introduce $|||q|||_p$ and not use it in Lemma 2.1?

* Lemma 2.2, neither $\iota$ nor $\Sigma$ have been defined. The lemma’s relevance relies in these definitions.

* Lemma 3.2, "with C$^*$ is a constant determined by Poincaré inequality", should be corrected for grammar.

* Final paragraph of Section 3, Sec. A should be Appendix A.

* Section 4, “and the introduction of stochastic gradient estimation then leads to new difficulties in another aspect.”, could be sharpened.

* In light of the many interesting theoretical results presented in this work, the conclusion (Section 5) feels rather abrupt. Perhaps the authors could develop this a bit further. Also, more attention should be paid to grammar, see, e.g., the final sentence.

---

> ### Author Response · Authors · 2026-06-12
>
> We thank the reviewer for their careful reading and very insightful comments on our work. We provide detailed responses to each of the requested changes. We will also provide an updated article that reflects all the changes we describe here once we have had a chance to respond to the feedback from all the reviewers, when the official response period is opened up. Please bear with us till all the reviews are be posted.
>
> Response to Questions:
> * The paper referred to is the following, and we will include it in the reference:
>      J. Cheeger, A lower bound for the smallest eigenvalue of the Laplacian, *Problems in Analysis (R. C. Gunning, ed.)*, Princeton Univ. Press (1970), 195–199.
> * We thank the reviewer for pointing out the need to be more precise here. We will modify the statement to :
> "Ghosh et al. (2025a) describe a technique of using multi-modal Gaussian mixtures as the kinetic energy $V(p)$ in order to help accelerate the convergence of HMC. They construct the mixture adaptively by placing modes at the observed cluster points that emerge from the HMC iterates, which are likely locations of modes of the potential function $U(q)$. The empirically observed acceleration is implied to be because the $V(p)$ adaptively adopts the multi-modal characteristics of $U(q)$."
> * Thank you for pointing this out, we will change it to *noisy gradient estimation*, which is used in stochastic gradient HMC literature. See e.g. Chen, Tianqi, Emily Fox, and Carlos Guestrin. *Stochastic Gradient Hamiltonian Monte Carlo,* International conference on machine learning. PMLR, 2014.
> * It should be “global minima of $U(q)$”.
> * Thank you for pointing this out. It is a typo, the correct statement should be " ... there exist positive constants $C_1$, $C_2$, $C^\ast$ and $\eta^\ast$, where $C^\ast$ is the constant appearing in Poincaré’s inequality for general measures .."
> * The threshold for step size $\eta$, $\eta^*$ is determined by $C_1$ and $C_2$. While we included a statement in the Theorem, we did not make the dependence explicit, we will provide the precise form of the dependence in the revision.
> * We will rephrase to make the sentences more concrete: "Further precise characterization of the rate requires improved quantification of the constants involved in functional inequalities such as the Poincaré's inequality, which are of wide general interest to the literature."
> * These are generalizations of results that were obtained in previous work under the assumption of Gaussian auxiliary distribution. These two lemmata thus allow the approaches in the previous works to be generated to include other auxiliary distributions. You are correct in that these are not lemmata used in the proofs of our propositions and theorems, so we will change them to propositions that are of independent interest, and clarify their roles more clearly in the introduction.

---

> > ### Author Response · Authors · 2026-06-12
> >
> > Response to Comments:
> >
> > * We will add an example on sampling from the Bayesian posterior of the parameters of a statistical learning exercise, where the likelihood model is defined and evaluated over a large dataset of samples.
> > * We thank  the reviewer for this suggestion, and we will modify the introduction accordingly.
> > * We will correct the grammar here.
> > * We will correct the grammar here.
> > * It is explicitly used in the proofs in the appendix. The expression in Lemma 2.1 is in fact a form of the triple norm, but we write it in the current form to specify the distributions involved, and we will add a remark to clarify that.
> > *  We regret the oversight and will add the definitions.
> > * We will correct the grammar here.
> > * We will correct the typo here.
> > * We will correct the grammar and sharpen the text here.
> > * We will revise the conclusions to say the following:
> >
> > In this paper, we analyzed the convergence of the HMC method where (a) the gradient is accessed only via a noisy estimator (that is, as a Stochastic Gradient), (b) Hamiltonian motion is discretized with leapfrog steps and (c) the auxiliary takes a general form that requires HMC to be modified to take steps with alternating directions. We take an analytic method approach to our investigation and derive bounds on the geometric convergence rates for a large family of SGHMC algorithms with general auxiliary distributions.
> >
> > As more applications of HMC emerge from different areas of machine learning, we expect these results to allow the presented algorithms to be adapted more readily and with higher confidence. We also expect the analytic methods developed in the paper can be more extensively utilized in the analysis of algorithms in this domain.
> >
> > For future research, we would like to explore HMC algorithms on Riemannian manifold and path spaces. For HMC on Riemannian manifold, see, e.g. Girolami et al (2011), the dependence of auxiliary distribution on the state variable poses new challenge to its convergence analysis. Meanwhile, HMC on path spaces, which are infinite dimensional, such as those proposed in Pinski (2021), requires new methodologies for its understanding. We expect that the Dirichlet form based method developed in this paper to be key elements for quantitative analysis of these algorithms.

---

> ### Comment · Reviewer_xgZe · 2026-06-16
>
> I thank the authors for their response to my questions and comments.
>
> Since the concerns raised by the other reviewers around Assumption 1 seem highly relevant, I am curious to see the authors' response to these. In particular, it seems that the "uniform strongly logarithmic concavity" assumption is required not only for the auxiliary, but also for the target. This seems very limiting, as the reason we use HMC is precisely to explore highly complex targets, which likely do not satisfy Assumption 1. Am I missing something here?

---

> > ### Comment · Reviewer_xgZe · 2026-07-15
> >
> > Could the authors please clarify if the target distribution also needs to satisfy Assumption 1? As stated above, this would be quite limiting.

---

> > > ### Author Response · Authors · 2026-07-16
> > >
> > > As we have observed in our response to reviewer 8FyC: In Assumptions 1 and 2, we identify a class of density functions for both target and auxiliary distributions. The conditions imposed on the target distribution are similar to many considered in the literature, for example Chen & Gatmiry (2023). Since we do not require Cheeger’s isoperimetric inequality to hold, the function class satisfying Assumptions 1 & 2 is less restrictive than those imposed in the literature. It is pertinent to note that we are able to conduct convergence analysis for ADHMC with auxiliary distributions drawn from the same function class as the target, unlike the most existing work whose auxiliary distribution is restricted to Gaussian.

---

### Review · Reviewer_8FyC · 2026-06-14

**Summary Of Contributions:**

the paper aims to derive quantitative  convergence rates for a family of HMC algorithms in which the Hamiltonian flow is approximated by a leapfrog integrator with MH correction, the potential gradient is come from  an unbiased stochastic gradient oracle, and the auxiliary momentum distribution is claimed to be general rather than gaussian. The authors’ knowledge claim these are the first convergence-rate results for HMC at this level of generality.

The analysis rests on

1.  a third-order analysis of the leapfrog map (Lemmas 4.1–4.2 and Appendix A) yields bounds of thedeviation from the exact flow at $(O(\eta^3)$ per step
2.  a Dirichlet-form / functional-inequality argument (Lemmas 3.1–3.2) which turns that integrator accuracy into a lower bound  decay in total variation (Theorem 3.1, Corollary 2).
3.  a structural “derivative condition” on the auxiliary (Assumption 4)  that  makes the argument go through, and is shown to hold for multivariate Gaussians and mixtures of Gaussians (Proposition 3.1, Corollary 1)

Strenghts

-  The Dirichlet-form route appears principled, and the leapfrog error bounds are detailed and reusable beyond this paper.
- Assumption 4 is a clean, interpretable condition, and its closure under Gaussian mixtures (Proposition 3.1) is a tidy result.

Weaknesses:

Unless I misunderstood something, the paper overclaims its contributions (see next section)

**Audience:**

Yes

**Audience Explanation:**

Convergence analysis of HMC is an active line within the MCMC and sampling-theory community, and this paper sits squarely in it.

**Claims And Evidence:**

No

**Claims Explanation:**

1. I might have misunderstood something, but I think the authors assumption 1 implies unimodality of the axuilliary (via strong log concavity implied by it?), so the "general" or "abitrary" contribution is threatened by this? I couldn't find any section of the paper which addresses this ( I think any difference of means >= 2 sigma would break the assumption?)
2. similarly, lemma 2 implicitly assumes symmetry (and I couldn't find any refernce to it later on, although it seems to be implicitly used) and uncorrelated/centered momentum?
3. for theorem 3.1, I'm not sure reversibility of the stochastic gradient induced operator is established in the paper? there's also an implicit warm start condition which I think an earlier section notes as a requirement

**Requested Changes:**

please correct me if I have misunderstood anything (and maybe reorganize to make it easier to understand the argumets)

also, I believe page 23 has a missing square on the integrand

---

> ### Author Response · Authors · 2026-06-26
>
> We thank the reviewer for their careful reading of our paper and for the constructive comments and suggestions. We have modified our manuscript as explained below, and we will shortly upload the revised version.
>
> **Q1**: We agree that our result statements should be presented more accurately. In Assumptions 1 and 2, we identify a class of density functions for both target and auxiliary distributions. The conditions imposed on the target distribution are similar to many considered in the literature, for example *Chen \& Gatmiry* (**2023**). Since we do not require Cheeger’s isoperimetric inequality to hold, the function class satisfying Assumptions  1 \& 2 is less restrictive than those imposed in the literature. It is pertinent to note that we are able to conduct convergence analysis for ADHMC with auxiliary distributions drawn from the same function class as the target, unlike the most existing work whose auxiliary distribution is restricted to Gaussian.
>
> Please note that our assumptions are also comparable to those in *Stoltz \& Trstanova* (**2018**), where the analysis of HMC is extended to a general class of symmetric auxiliary distributions under smoothness assumptions similar to ours. Moreover, the convergence analysis in *Stoltz \& Trstanova* (**2018**) were limited to an idealized continuous time flow abstraction, not a practical implementation of HMC.
>
> We have implemented the following changes in the manuscript in response to this question:
>
> * In the abstract, we have stated that "our analysis extends the class of auxiliary distributions allowed via a novel HMC procedure of alternating directions (AD)".
>
> * Corresponding statements in the main text have been modified accordingly.
>
> * We have added a brief discussion of the points above in the appropriate places in the literature review section.
>
> **Q2**: Lemma 2.2 has been removed in response to a reviewer comment. Our results actually do not require symmetry.
>
> **Q3**: We have clarified that Theorem 3 applies to ADHMC. This algorithm's reversibility is established in *Ghosh et al* (**2025**) , as has been pointed out at the end of Sec. 2.3.3.
>
> **Requested Changes**: We have applied the suggested corrections to the revised paper.

---

### Review · Reviewer_W6e3 · 2026-06-22

**Summary Of Contributions:**

The paper provides a convergence proof of Alternating Directions Stochastic Gradient Descent (ADHMC) under various assumptions.

**Additional Comments:**

It might be worthwhile to combine the results LEM3.2 - COR 2 together for brevity, but this is a stylistic choice I leave to the authors.

**Audience:**

Yes

**Audience Explanation:**

Convergence proofs of variants of HMC algorithms are of critical value to the ML community.

**Broader Impact Concerns:**

None.

**Claims And Evidence:**

Yes

**Claims Explanation:**

The entirety of the paper deals with providing the convergence proof, which results in a variant

**Requested Changes:**

Apologies for the very late review, and I will ask only for a couple changes, in the hope the changes can improve the manuscript.

The convergence estimates in THM 3.1 and COR 2 (strange numbering for the corollary, by the way) appear to differ only in the constants, $A_3$ versus $A_3^{SG}$ given in LEM 4.5-4.6. It would be helpful to the reader if the authors would parse the differences in these constants, and briefly discuss the implications for the differences.

I got the impression from the introduction that THM 3.1 is perhaps already known? If yes, please add a reference, if it is new perhaps add titles to both THM 3.1 and COR 2, which appear to be the main results the readers should focus on.

- In DEF 2, I presume the last inequality is required for all $u \in L^2(X,m)$?
- This Dirichlet form and the conditions LEM 3.1 appear very central but the comments appearing before LEM 3.1 seems repetitive... can the comments be updates to offer more insight into this condition?

---

> ### Author Response · Authors · 2026-06-26
>
> We thank the reviewer for their careful reading of our paper and for the constructive comments and suggestions. We have modified our manuscript as explained below, and we will shortly upload the revised version.
>
> **Theorem 3.1** is a new contribution of this paper since it analyses the ADHMC algorithm (As we have clarified in its statement). It covers the case where gradient computation is exact and the Hamiltonian motion is implemented via symplectic discretization. We separate the additional impact of stochastic gradient estimation to Corollary 2 (Corollary 3.2 in the latest version pursuant to your naming suggestion) to emphasize the different impacts each have. We have modified the introduction to include both of them in the contributions section, and we have added a brief  discussion in the paragraph below Corollary 3.2 on the implication of the difference in the key constants.
>
> **Defn 2.1** You are correct, and we have added the needed quantifier to the statement.
>
> **Dirichlet form discussion** We have reorganized and modified the first several paragraphs of Sec. 3 to improve its readability.

---

### Author Response · Authors · 2026-06-26
**Response to reviewers **xgZe** and **WjEg** on Asymmetric Auxiliary Example**

We thank the reviewers for raising this question on the need for a concrete auxiliary distribution example that is asymmetric and  satisfies both Assumptions 1 \& 4. We have added a new one-dimensional Example 1 after presenting Assumption 4 and Corollary 3.1. It is formed by "stitching" together three Gaussians in a manner that renders a non-symmetric distribution  satisfying Assumptions 1 \& 4.

We have also added a remark in the conclusion: "We would like to complete an analysis of ADHMC convergence by relaxing the Lipschitz-smoothness condition on $\nabla V$ in Assumption 1 to allow for mixtures of Gaussians." In particular, mixtures of Gaussians satisfy a relaxed condition known as Lipschitz-upper-smoothness for some $L>0$:
$$
V(q)\leq V(q_0)+\left\langle \nabla V(q_0),q-q_0\right\rangle +\frac{L}{2}\left\Vert q-q_0\right\Vert ^{2},\quad\forall q, q_0\in {\mathbb{R}}^{d}.
$$
Analysis of convergence under this condition is significantly harder and will be the subject of our folllow-on work.

---

### Author Response · Authors · 2026-06-26
**Revised Manuscript Uploaded**

Please find the revised version of the manuscript incorporating all the changes discussed below with the reviewers now attached to this submission. We thank all the reviewers again in contributing to strengthen the manuscript.

(Our apologies for the multiple comments added today, which have consequently generated multiple email notifications.)